# Multiplexed proteomics of autophagy-deficient murine macrophages reveals enhanced antimicrobial immunity via the oxidative stress response

Timurs Maculins[1,2], Erik Verschueren[3†], Trent Hinkle[3], Meena Choi[3,4], Patrick Chang[5], Cecile Chalouni[5], Shilpa Rao[6], Youngsu Kwon[7], Junghyun Lim[1], Anand Kumar Katakam[5], Ryan C Kunz[8], Brian K Erickson[8], Ting Huang[4], Tsung-Heng Tsai[4,9], Olga Vitek[4], Mike Reichelt[5], Yasin Senbabaoglu[6], Brent Mckenzie[7], John R Rohde[10], Ivan Dikic[2,11]*, Donald S Kirkpatrick[12]*, Aditya Murthy[12]*

[1]Department of Cancer Immunology, Genentech, South San Francisco, United States; [2]Institute of Biochemistry II, Goethe University, Frankfurt am Main, Germany; [3]Department of Microchemistry, Proteomics and Lipidomics, Genentech, South San Francisco, United States; [4]Khoury College of Computer Sciences, Northeastern University, Boston, United States; [5]Department of Pathology, Genentech, South San Francisco, United States; [6]Department of Oncology Bioinformatics, Genentech, South San Francisco, United States; [7]Department of Translational Immunology, Genentech, South San Francisco, United States; [8]IQ Proteomics LLC, Cambridge, United States; [9]Department of Mathematical Sciences, Kent State University, Kent, United States; [10]Department of Microbiology and Immunology, Dalhousie University, Halifax, Canada; [11]Department of Infectious Diseases, Genentech, South San Francisco, United States; [12]Interline Therapeutics, South San Francisco, United States

*For correspondence:
dikic@biochem2.uni-frankfurt.de (ID);
dkirkpatrick@interlinetx.com (DSK);
amurthy@interlinetx.com (AM)

Present address: †Galapagos NV, Mechelen, Belgium

**Abstract** Defective autophagy is strongly associated with chronic inflammation. Loss-of-function of the core autophagy gene *Atg16l1* increases risk for Crohn's disease in part by enhancing innate immunity through myeloid cells such as macrophages. However, autophagy is also recognized as a mechanism for clearance of certain intracellular pathogens. These divergent observations prompted a re-evaluation of ATG16L1 in innate antimicrobial immunity. In this study, we found that loss of *Atg16l1* in myeloid cells enhanced the killing of virulent *Shigella flexneri* (*S.flexneri*), a clinically relevant enteric bacterium that resides within the cytosol by escaping from membrane-bound compartments. Quantitative multiplexed proteomics of murine bone marrow-derived macrophages revealed that ATG16L1 deficiency significantly upregulated proteins involved in the glutathione-mediated antioxidant response to compensate for elevated oxidative stress, which simultaneously promoted *S.flexneri* killing. Consistent with this, myeloid-specific deletion of *Atg16l1* in mice accelerated bacterial clearance in *vitro* and *in vivo*. Pharmacological induction of oxidative stress through suppression of cysteine import enhanced microbial clearance by macrophages. Conversely, antioxidant treatment of macrophages permitted *S.flexneri* proliferation. These findings demonstrate that control of oxidative stress by ATG16L1 and autophagy regulates antimicrobial immunity against intracellular pathogens.

## Introduction

Effective immunity against enteric pathogens requires complex signaling to coordinate the inflammatory response, pathogen clearance, tissue remodeling, and repair (*Maloy and Powrie, 2011*). Autophagy, a cellular catabolic pathway that eliminates cytosolic cargo via lysosomal degradation, has emerged as an important regulator of mucosal immunity and inflammatory bowel disease (IBD) etiology. Genome-wide association studies linked the missense T300A variant in the core autophagy gene *Atg16l1* with increased risk for IBD (*Hampe et al., 2007*; *Rioux et al., 2007*). Later studies demonstrated that this variant contributes to enhanced caspase-mediated degradation of the ATG16L1 protein (*Lassen et al., 2014*; *Murthy et al., 2014*). Genetic loss-of-function of core autophagy genes including *Atg16l1* increases secretion of pro-inflammatory cytokines by macrophages in response to toll-like receptor (TLR) activation (*Lim et al., 2019*; *Saitoh et al., 2008*). This contributes to increased mucosal inflammation, driving resistance to extracellular pathogens such as *Citrobacter rodentium* (*Marchiando et al., 2013*; *Martin et al., 2018*) and pathogenic *Escherichia coli* (*Wang et al., 2019*). Loss of autophagy-related genes *Atg16l1*, *Beclin1*, and *Rbc1cc1* (encoding FIP200) in the myeloid compartment also confers enhanced antimicrobial immunity against certain intracellular pathogens, such as *Salmonella typhimurium* (*S.typhimurium)* and *Listeria monocytogenes* via induction of type I and II interferon responses (*Samie et al., 2018*; *Wang et al., 2020*). Thus, autophagy impacts antimicrobial immunity *in vivo* via innate immunosuppression as well as myeloid cell reprogramming.

Targeted elimination of intracellular pathogens by xenophagy, a form of selective autophagy, is well-described in cellular model systems (*Bauckman et al., 2015*). In contrast to non-selective autophagy triggered by nutrient stress, xenophagy functions to eliminate intracellular bacteria by sequestering them in autophagosomes and shuttling them to the degradative lysosomal compartment. Pathogenic bacteria have evolved mechanisms to either evade capture by the autophagy machinery, as by *S.typhimurium* and *S.flexneri* (*Birmingham et al., 2006*; *Campbell-Valois et al., 2015*; *Dong et al., 2012*; *Martin et al., 2018*; *Xu et al., 2019b*) or attenuate autophagic flux as by *Legionella pneumophila* (*Choy et al., 2012*). *S.typhimurium* primarily resides in a protective compartment known as the *Salmonella* containing vacuole (SCV). There it prevents formation of the ATG5~ATG12-ATG16L1 complex at the bacterial vacuolar membrane via secretion of the effector SopF, which blocks ATG16L1 association with vacuolar ATPases (*Xu et al., 2019b*). Despite its ability to interfere with autophagy, infected host cells still recognize 10–20% of cytosolic *S.typhimurium* and subject this subpopulation to lysosomal degradation via mechanisms involving direct recognition of either the bacterial surface (*Huang and Brumell, 2014*; *Stolz et al., 2014*) or damaged phagocytic membranes (*Fujita et al., 2013*; *Thurston et al., 2012*).

Compared to *S.typhimurium*, *S.flexneri* is not characterized by a vacuolar life cycle, but instead resides in the host cytoplasm. *S.flexneri* effector proteins IcsB and VirA are capable of completely inhibiting autophagic recognition to permit replication in the host cytosol (*Liu et al., 2018*; *Ogawa et al., 2005*). In response, the host cell attempts to further counteract *S.flexneri* infection via diverse mechanisms, such as coating bacterial cell surfaces with guanylate-binding proteins (GBPs) (*Li et al., 2017*; *Wandel et al., 2017*) or sequestering bacteria in Septin cage-like structures to restrict their motility (*Mostowy et al., 2010*). To reveal these mechanisms, cell-based studies have largely utilized attenuated variants (e.g. IcsB or IcsB/VirA double mutants of *S.flexneri*) or strains that inefficiently colonize the host cytosol (e.g. *S.typhimurium* which express SopF). Thus, observations from *in vivo* genetic models must be reconciled with observations made in cell-based systems to fully describe the roles of autophagy in antimicrobial immunity. Importantly, there is a lack of understanding of how autophagy contributes to immunity against non-attenuated *S.flexneri*. This insight is especially lacking in relevant host cell types, such as macrophages that are infected by *S.flexneri* (*Ashida et al., 2015*).

In this study, we investigated the role of ATG16L1 in myeloid cells in response to infection by virulent *S.flexneri* (strain M90T). We observed that loss of *Atg16l1* in murine bone-marrow-derived macrophages (BMDMs) enhanced *S.flexneri* elimination in cell culture, as well as by mice lacking ATG16L1 in the myeloid compartment (Atg16l1-cKO; LysM-*Cre*-mediated deletion of *Atg16l1*). We utilized multiplexed quantitative proteomics to characterize total protein, phosphorylation, and ubiquitination changes in wild type (WT) and ATG16L1-deficient (cKO) BMDMs either uninfected or infected with *S.flexneri*. Together these data provide a comprehensive catalogue of basal differences

between WT and cKO BMDMs and the dynamic response of each to infection. Expectedly, significant differences were observed for components in the autophagy pathway, along with proteins involved in cell death, innate immune sensing and NF-κB signaling. However, pharmacological inhibition of these pathways only modestly affected *S.flexneri* killing by BMDMs at later time points following infection. One key difference between control and ATG16L1-deficient BMDMs was a basal oxidative stress response, highlighted by accumulation of the sodium-independent cystine-glutamate antiporter (XCT) and multiple components of the glutathione metabolic pathway. XCT is responsible for importing the constituents required for glutathione (GSH) biosynthesis, and the consequent detoxification of reactive oxygen species (ROS) and lipid peroxides. Consistently, sustained pharmacological XCT inhibition in ATG16L1-deficient BMDMs compromised their viability. Likewise, XCT inhibition enhanced *S.flexneri* clearance by wild type BMDMs, demonstrating a role for this pathway in eliminating cytosolic bacteria. Finally, pharmacological scavenging of ROS permitted bacterial regrowth in BMDMs, thereby linking the antimicrobial capacity of ATG16L1-deficient macrophages to elevated ROS.

This study offers a comprehensive, multidimensional catalogue of proteome-wide changes in macrophages following infection by an enteric cytosolic pathogen, including key nodes of cell-autonomous immunity regulated by autophagy. Our findings demonstrate that ATG16L1 tunes antimicrobial immunity against cytosolic pathogens via oxidative stress as well as interferon responses. Pharmacological modulation of these pathways may represent opportunities for elimination of cytosolic pathogens.

## Results

### Enhanced clearance of intracellular *S.flexneri* by loss of *Atg16l1*

Recent studies identified that defective autophagy in macrophages enhances type I interferon and TNF-driven inflammatory signaling to promote antimicrobial immunity (*Lim et al., 2019*; *Martin et al., 2018*; *Samie et al., 2018*; *Wang et al., 2020*). Given these observations, we wanted to explore whether loss of *Atg16l1* affects killing of the intracellular enteric pathogen *S.flexneri* (strain M90T). To test this directly in BMDMs, cells from either control (*Atg16l1*-WT) or mice lacking ATG16L1 in the myeloid compartment (*Atg16l1*-cKO) were subjected to the gentamycin protection assay that enables quantification of intracellular bacteria by enumerating colony forming units (CFUs). We first determined the kinetics of *S.flexneri* killing by following BMDM infection in a time-course experiment with multiplicity of infection (MOI) of 5. Compared to wild type (WT) controls, ATG16L1-deficient BMDMs (cKO) demonstrated enhanced bacterial clearance (*Figure 1A and B* and *Figure 1—figure supplement 1A*). Since prior studies demonstrated increased sensitivity of autophagy-deficient cells to programmed cell death following engagement of cytokine receptors or microbial ligands (*Lim et al., 2019*; *Matsuzawa-Ishimoto et al., 2017*; *Orvedahl et al., 2019*), BMDM viability was measured in parallel by quantifying the propidium iodide (PI)-positive population via live-cell imaging. WT and cKO BMDMs displayed similar cell death kinetics over the time course of infection, indicating that the phenotype was not driven by enhanced cell death, but potentially by other factors in cKO BMDMs (*Figure 1C*).

We next asked whether ATG16L1 in myeloid cells similarly impacts antimicrobial immunity *in vivo*. Since chronic enteric infection with *S.flexneri* in wild-type mice is not possible, current murine models are limited to acute infection via intravenous or intraperitoneal routes. We performed tail vein injection of *S.flexneri* and evaluated bacterial colonization in liver, spleen, and lung tissues at 6 or 24 hr following infection. Infection of the lung was not detected (data not shown) and infection of the spleen was comparable between genotypes (*Figure 1—figure supplement 1B*). However, myeloid-specific loss of *Atg16l1* resulted in a markedly decreased bacterial burden in the liver (*Figures 1D*, 6 hr; *Figures 1E*, 24 hr). These findings indicate that ATG16L1 deficiency in macrophages enhances clearance of intracellular *S.flexneri in vitro*, while myeloid-specific loss of *Atg16l1* accelerates hepatic clearance of *S.flexneri in vivo*. Although enhanced clearance of *S.flexneri* by *Atg16l1*-cKO mice at the early time point (6 hr) supported a role for ATG16L1 in hepatic myeloid cells including macrophages, a Kupffer-cell-specific role cannot be asserted using the current model since the LysM-*cre* transgene induces *Atg16l1*-cKO in other myeloid cell populations that may also contribute to the *in vivo* phenotype.

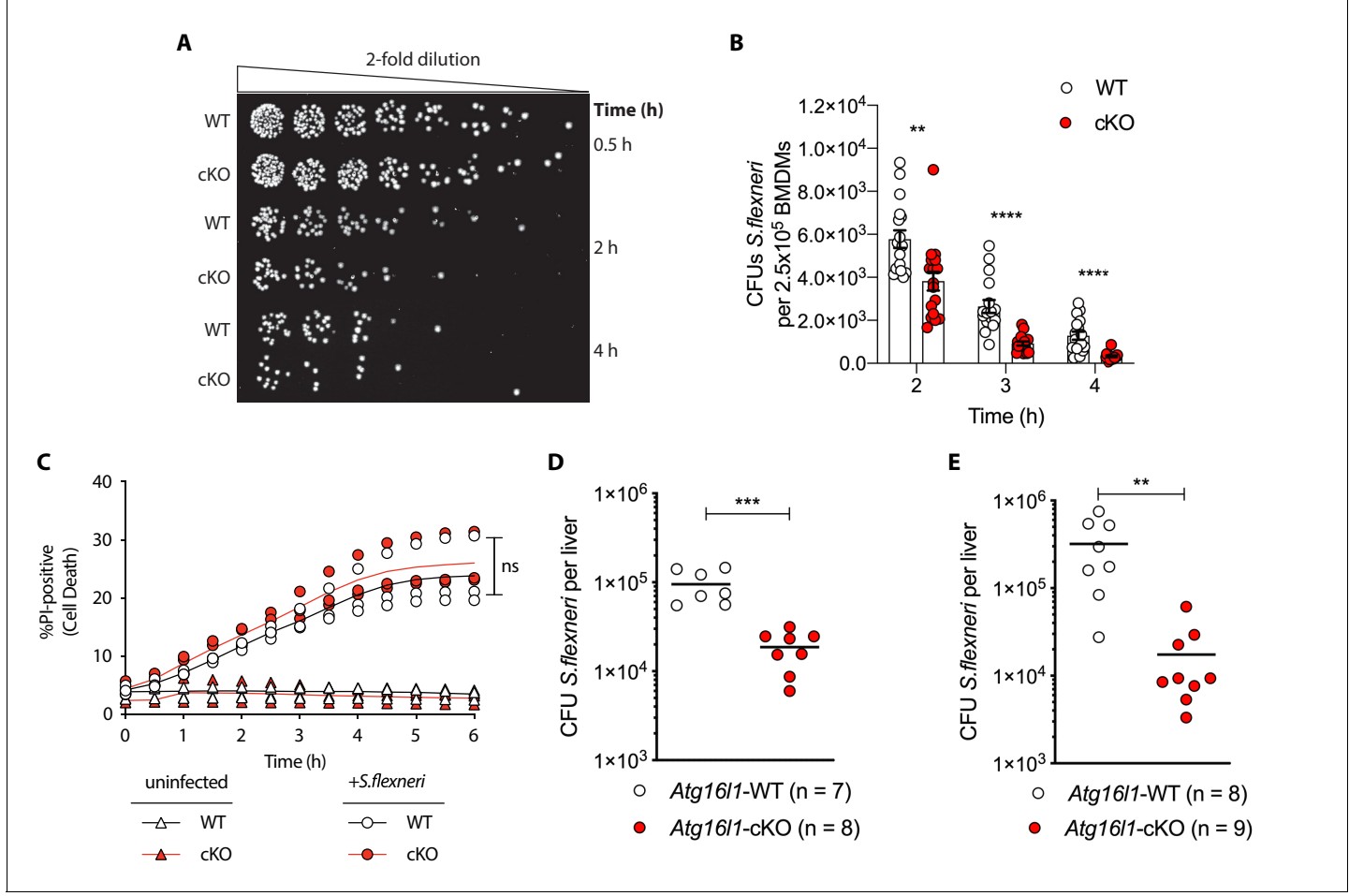

**Figure 1.** Enhanced clearance of intracellular *S.flexneri* by loss of *Atg16l1*. (**A**) Representative serial dilutions from gentamycin protection assays following *S.flexneri* M90T infection of WT or cKO BMDMs at the indicated timepoints. (**B**) Comparison of colony forming units (CFUs) per well from independent infection experiments using BMDM preparations from *Atg16l1*-WT or *Atg16l1*-cKO mice. 2 hr **p=0.002, 3 hr **** p = <0.0001, 4 hr **** p = <0.0001, multiple t-test comparison. (**C**) Percentage of propidium iodide (PI)-positive cells during time-course infection of WT or cKO BMDMs with *S.flexneri* M90T. Graph represents individual values from three independent experiments using three different BMDM preparations. ns, non-significant. (**D, E**) Liver bacterial load 6 hr (**D**) or 24 hr (**E**) following intravenous injection of *Atg16l1*-WT or *Atg16l1*-cKO mice with *S.flexneri* M90T. Graphs show data from representative experiments out of two (**D**) or four (**E**) independent experiments as log10 CFU count per liver in indicated number of mice. In (**D**) ***p=0.0002 and in (**E**) outliers removed using ROUT (Q = 1%) method, **p=0.0031.

The online version of this article includes the following figure supplement(s) for figure 1:

**Figure supplement 1.** Enhanced clearance of *S.flexneri* by loss of *Atg16l1*.

## Multiplexed proteomic profiling of BMDMs following infection

To reveal factors that may drive enhanced *S.flexneri* killing in ATG16L1-deficient BMDMs, we characterized changes in the global proteome and post-translational modifications (PTMs) between WT and cKO BMDMs. Specifically, we applied 11-plex isobaric multiplexing via tandem mass tagging (TMT) in combination with liquid-chromatography and tandem mass spectrometry (LC-MS/MS). Cell lysates were prepared from WT and cKO BMDMs that were either **U**ninfected (U) or infected with *S.flexneri* (MOI 5) at **E**arly (E; 45–60 min) or **L**ate (L; 3–3.5 hr) time-points (*Figure 2A*). Cumulatively, two 11-plex experiments were performed with uninfected samples represented in biological triplicates and infected samples represented in biological quadruplicates (see Materials and methods for details). Data were acquired using the synchronous precursor selection (SPS)-MS3 approach wherein dedicated MS3 scan events are collected from fragment ion populations representing a mixture of the 11 samples and used to report the relative abundance of each peptide feature per channel (*McAlister et al., 2014*; *Ting et al., 2011*; *Figure 2A*).

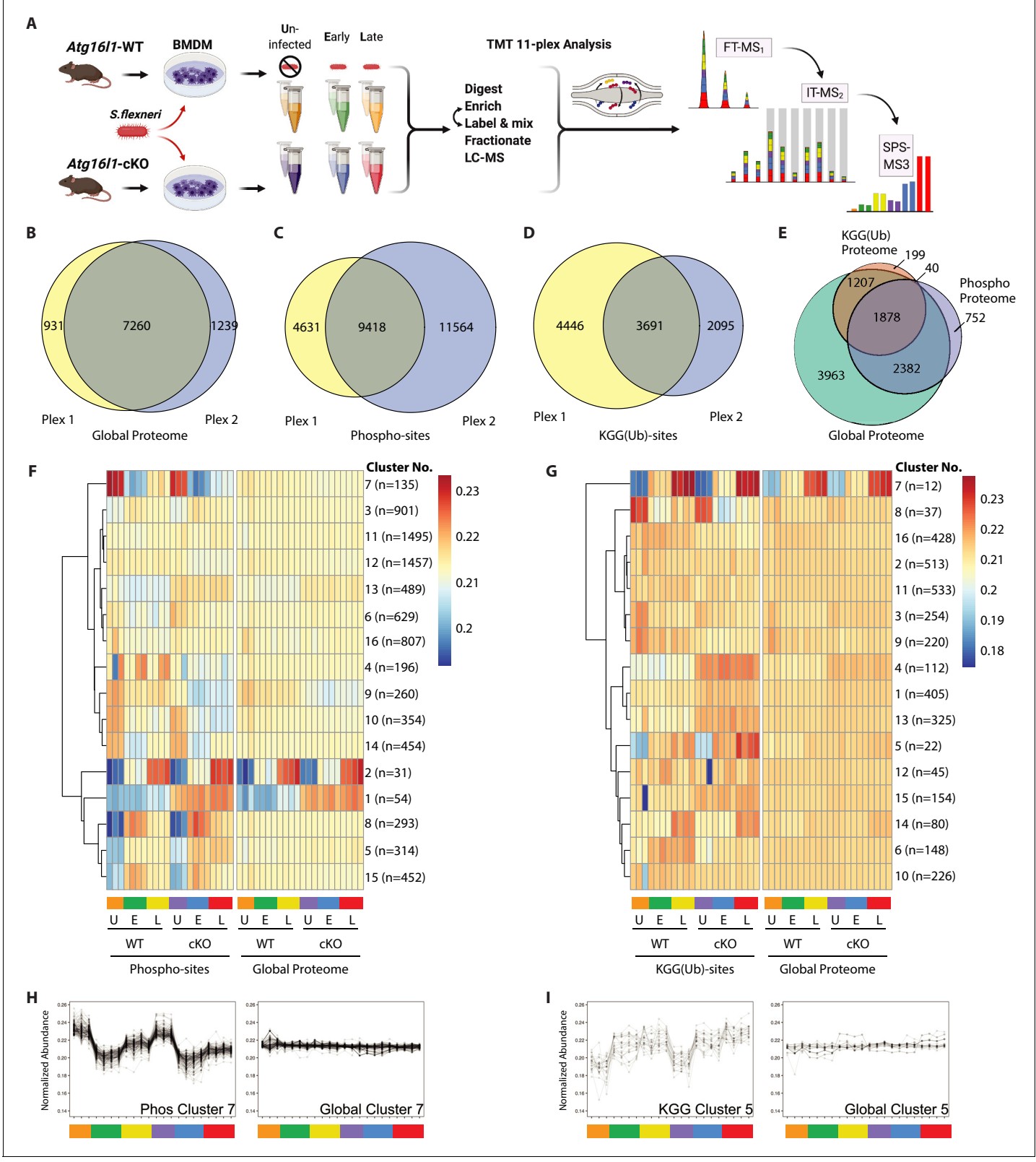

**Figure 2.** Multiplexed proteomic profiling of BMDMs following infection. (**A**) Schematic representation of multiplexed proteomic profiling of macrophages during *S.flexneri* infection. (**B–D**) Venn diagrams show overlapping quantitative data collected in Plex1 and/or Plex2 for (**B**) global proteins, (**C**) phospho-, and (**D**) KGG(Ub)-sites. (**E**) Venn diagram displays an overlap of quantitative data for phospho- and KGG(Ub)-sites with respect to the global proteins quantified. (**F and G**) Heatmaps displaying K-means clustered quantitative data for (**F**) phospho- and (**G**) KGG(Ub)-sites relative to

*Figure 2 continued on next page*

*Figure 2 continued*

their corresponding global proteins. Note that the global proteins subjected to clustering differ between panels F and G based on the proteins from which PTMs were quantified. (**H, I**) Line plots show representative clusters from the Heatmaps above. Phospho Cluster 7 (panel H) and KGG(Ub) Cluster 5 (panel I) each show PTM profiles that diverge from their corresponding global protein measurements. Proteins and PTMs making up each cluster are presented in *Supplementary file 1*.

The online version of this article includes the following figure supplement(s) for figure 2:

**Figure supplement 1.** Quality control and PTM-Global comparative analysis of proteomics data.

For global proteome profiling, quantitative data were obtained from >103,700 peptide features mapping to 9430 proteins. From the PTM enriched samples, quantitative data were obtained for >25,600 unique phospho- (5052 proteins) and >12,400 unique KGG(Ub)-sites (3324 proteins). When considering only features bearing data in both 11-plexes, the final dataset contained quantitative data for 7260 proteins (i.e. global proteome), 9418 phospho- and 3691 KGG(Ub)-sites (*Figure 2B–D*). As expected, ~90% of the post-translationally modified peptide spectral matches derived from proteins that were also identified and quantified in the global proteome dataset (*Figure 2E*). Both within and between plexes, peptide and protein level quantitative data were highly reproducible, with Pearson correlations ranging from 0.96 to 0.99 (*Figure 2—figure supplement 1A*). Phospho- and KGG(Ub)-sites profiling data were subjected to K-means clustering, each paired with the corresponding global proteome data. Heatmap representations revealed clusters of PTM changes that occur in genotype- and/or infection-dependent manners (*Figure 2F and G*). A subset of these clusters comprised PTMs whose quantitative profiles mirrored that of the underlying protein level due to altered protein expression or stability (e.g. phospho-sites Clusters 1–2 in *Figure 2F* and *Figure 2—figure supplement 1B*; KGG(Ub)-sites Cluster 7 in *Figure 2G* and *Figure 2—figure supplement 1C*). In contrast, other clusters displayed PTM profiles that diverged from their underlying proteins (e.g. phospho-sites Cluster 7 in *Figure 2F and H*; KGG(Ub)-sites Cluster 5 in *Figure 2G and I*). The composition of PTMs and proteins comprising each cluster are available in *Supplementary file 1*.

Interrogation of the uninfected datasets revealed differences between the genotypes on the global protein level. Consistent with previous observations (*Samie et al., 2018*), cKO BMDMs showed upregulation in autophagy receptors and inflammatory regulators, such as SQSTM/p62 and ZBP1, respectively (*Figure 3A*). In the phosphoproteome and KGG(Ub) datasets interesting observations concerned elevated phosphorylation of ubiquitin (RL40) at serine-57 (RL40_S57) (*Figure 3B*) and ubiquitination of FIS1 at lysine-20 (FIS1_K20) (*Figure 3C*), which are involved in endocytic trafficking (*Lee et al., 2017*; *Peng et al., 2003*) and mitochondrial and peroxisomal homeostasis (*Bingol et al., 2014*; *Koch et al., 2005*; *Zhang et al., 2012*), respectively.

Similar analysis of the infected datasets revealed the dynamic nature of the macrophage response to infection, irrespective of ATG16L1 genotype. For example, global proteome analysis revealed broad changes in pro-inflammatory cytokines and chemokines at early (GROA), late (CXL10, IL1A, IL1B) or both (CCL2, TNFA) time-points, as well as marked changes in several key cell surface receptors (*Figure 3D*, *Figure 3—figure supplement 1A and B*). Time-dependent changes were also observed for components of innate immune signaling that intersect with the ubiquitin pathway (PELI1/Pellino), kinase-phosphatase signaling (DUS1/Dusp1), and interferon-mediated GTP/GDP signaling (GBP5) (*Figure 3—figure supplement 1C*). For phosphorylation, notable examples included tyrosine-431 of the PI3-kinase regulatory subunit (P85A_Y431) which decreased rapidly upon infection in both genotypes, and S379 of the interferon regulatory factor (IRF3_S379) which was increased following infection, particularly in ATG16L1-deficient cells (*Figure 3E*). For ubiquitination, marked effects were seen for the selective autophagy receptor Tax1BP1 (TAXB1_K618) and an E3 ubiquitin ligase Pellino (PELI1_K202) (*Figure 3F*), both of which have defined roles at the intersection of cell death and innate immune signaling (*Choi et al., 2018*; *Gao et al., 2011*; *Parvatiyar et al., 2010*). Finally, annotation of the *S.flexneri* proteome permitted quantification of pathogen-derived proteins at the global level. These can be identified using the search term '*SHIFL' in the interactive dashboard provided below.

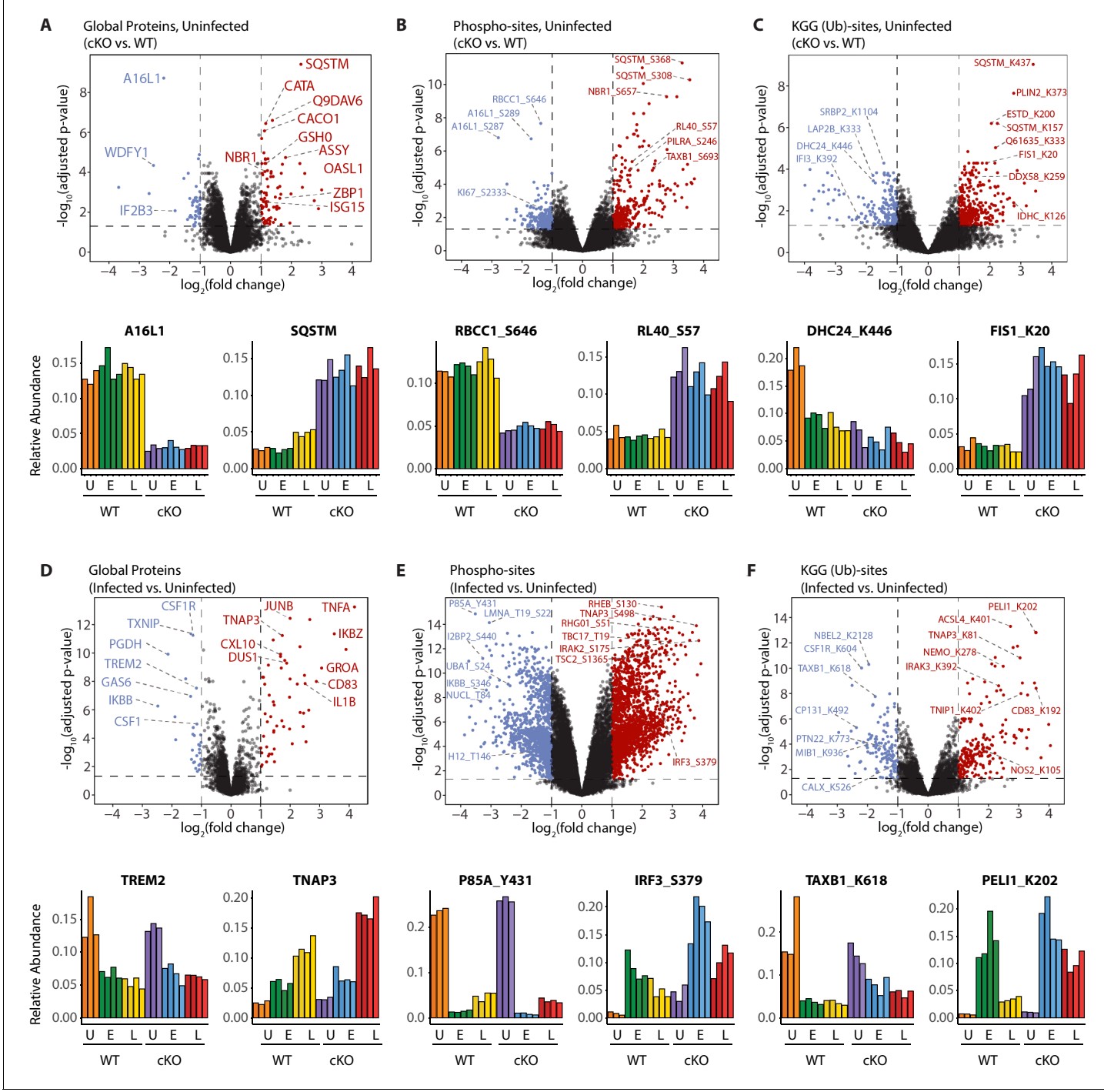

**Figure 3.** A global overview of changes identified between the genotypes and upon infection. (A–C) Volcano plots showing differential expression of global proteins (A), phospho- (B) and KGG(Ub)-sites (C) between uninfected cKO vs. WT BMDMs. Volcano plots display log2 fold changes and -log10 transformed adjusted p-values for the host proteome. Bar graphs at the bottom of each panel represent top hits with positive and negative log2 fold changes. Uninfected (U) samples are shown with orange (WT) and purple (cKO), early infection (E) in green (WT) and blue (cKO) and late infection (L) in yellow (WT) and red (cKO), respectively. Protein names are shown as UniProt identifiers with modification sites indicated by the modified amino acid (S/T/Y/K) and residue number (e.g. RL40_S57). Features enriched in cKO and WT BMDMs are highlighted in red and blue, respectively. (D–F) Volcano plots displaying differentially expressed global proteins (D), phospho- (E) and KGG(Ub)-sites (F) between infected and uninfected BMDMs. Infected refers to the aggregate condition in which early and late infected samples for WT and cKO are each weighted as 0.25 relative to 0.5 each for the WT and cKO uninfected samples. Features enriched in infected and uninfected BMDMs are highlighted in red and blue, respectively. As above, bar graphs below each panel show example hits. The relative abundance of TMT reporter ions sums up to 2.0 for features quantified in both Plex1 and Plex2.

*Figure 3 continued on next page*

*Figure 3 continued*

The online version of this article includes the following figure supplement(s) for figure 3:

**Figure supplement 1.** Dynamic macrophage response to infection.

To facilitate further exploration of pathways within these data, we have prepared interactive Spotfire dashboards and provided them as a resource to the community. These can be accessed at the following URL: https://info.perkinelmer.com/analytics-resource-center.

## Characterizing PTMs of autophagy proteins and inflammatory signaling nodes revealed by loss of *Atg16l1* and infection

To effectively integrate data for each protein within a single consolidated view, heatmaps were assembled to show proteome level changes alongside any PTMs that were quantified in the phospho- and KGG(Ub)-enriched samples. Using the selective autophagy receptor Tax1BP1 (TAXB1) as an example, heatmaps depict relative abundance of features present in one or both experiments (Plex1 and/or Plex2) (*Figure 4A*). Comparisons of interest include cKO versus WT (cKO/WT) for **U**ninfected (U), **E**arly (E) and **L**ate (L) infection time-points. For TAXB1, these show that the global protein level is elevated upon *Atg16l1* deletion, as are a number of individual phosphorylation and ubiquitination sites. These include features quantified in one (e.g. T494, K624) or both plexes (e.g. S632, S693, K627). Additional comparisons depict time-dependent differences between infected and uninfected conditions for each genotype – namely early versus uninfected (E/U) and late versus uninfected (L/U). For TAXB1, certain PTMs such as phosphorylation at S632 and ubiquitination at K624 and K627 track with the protein, while other PTMs such as phosphorylation at threonine-494 (T494) and S693 display time-dependent changes that diverge from the underlying protein level (*Figure 4A*). Shown individually, histograms depict the relative abundance of TAXB1 and its specific PTMs (*Figure 4B*) to mirror what is shown in the combined heatmaps (*Figure 4A*).

One pathway where we expected to see marked proteome and PTM level changes upon infection was in autophagy (*Figure 4C* and *Figure 4—figure supplement 1*). We confirmed genotype-dependent loss of each component of the ATG16L1-ATG5~ATG12 complex that conjugates LC3 (MLP3A) to phosphatidylethanolamine (*Figure 4D*). Only modest changes were seen in the core autophagy machinery following infection, with the most notable effects being differential phosphorylation of FIP200 (RBCC1), ATG2B, and VPS15/p150 (PI3R4) (*Figure 4—figure supplement 1C–E*). More substantial effects were seen for phosphorylation events on autophagy receptors such SQSTM/p62 and Optineurin (OPTN) (*Figure 4E*), in addition to TAXB1 (*Figure 4A*). In the case of SQSTM/p62, singly and multiply phosphorylated forms of T269, T271, T272, S275/6, S277 were elevated in ATG16L1-deficient macrophages. S28 phosphorylation of SQSTM/p62 was previously described to regulate activation of the antioxidant response (*Xu et al., 2019a*). We detected a substantial increase in basal SQSTM_S28 phosphorylation in cKO BMDMs, indicating that ATG16L1 deficiency may impact oxidative stress (*Figure 4—figure supplement 1F*). Previous work from several groups have demonstrated that TBK1 regulates OPTN and SQSTM through phosphorylation, although the sites quantified in this dataset differ from those previously characterized in detail (*Heo et al., 2015*; *Matsumoto et al., 2015*; *Richter et al., 2016*).

Our PTM datasets showed dynamic regulation for a range of inflammatory signaling components by infection as well as autophagy (*Figure 4—figure supplement 2*). For example, we detected ubiquitination on K278 of NEMO (*Figure 4—figure supplement 2F*), consistent with increased LUBAC activity (*Tokunaga et al., 2009*). Interestingly, the global proteome data reported a peptide with the sequence GGMQIFVK that is derived from linear polyubiquitin chains formed by the LUBAC complex (*Figure 3—figure supplement 1D*). This linear ubiquitin peptide was elevated upon infection in both WT and cKO BMDMs, further supporting increased E3 ubiquitin ligase activity of LUBAC. As noted above, TAXB1 phosphorylation was induced upon infection at a number of sites (*Figure 4A*). These changes in TAXB1 correlated with numerous elevated PTMs of the A20 (TNAP3) deubiquitinase, a protein whose anti-inflammatory activity modulates NF-κB signaling (*Figure 4—figure supplement 2C*). Phosphorylation at TAXB1 at S693 is important for the assembly of TNAP3-containing complex and negative regulation of NF-κB signaling (*Shembade et al., 2011*; *Figure 4A*).

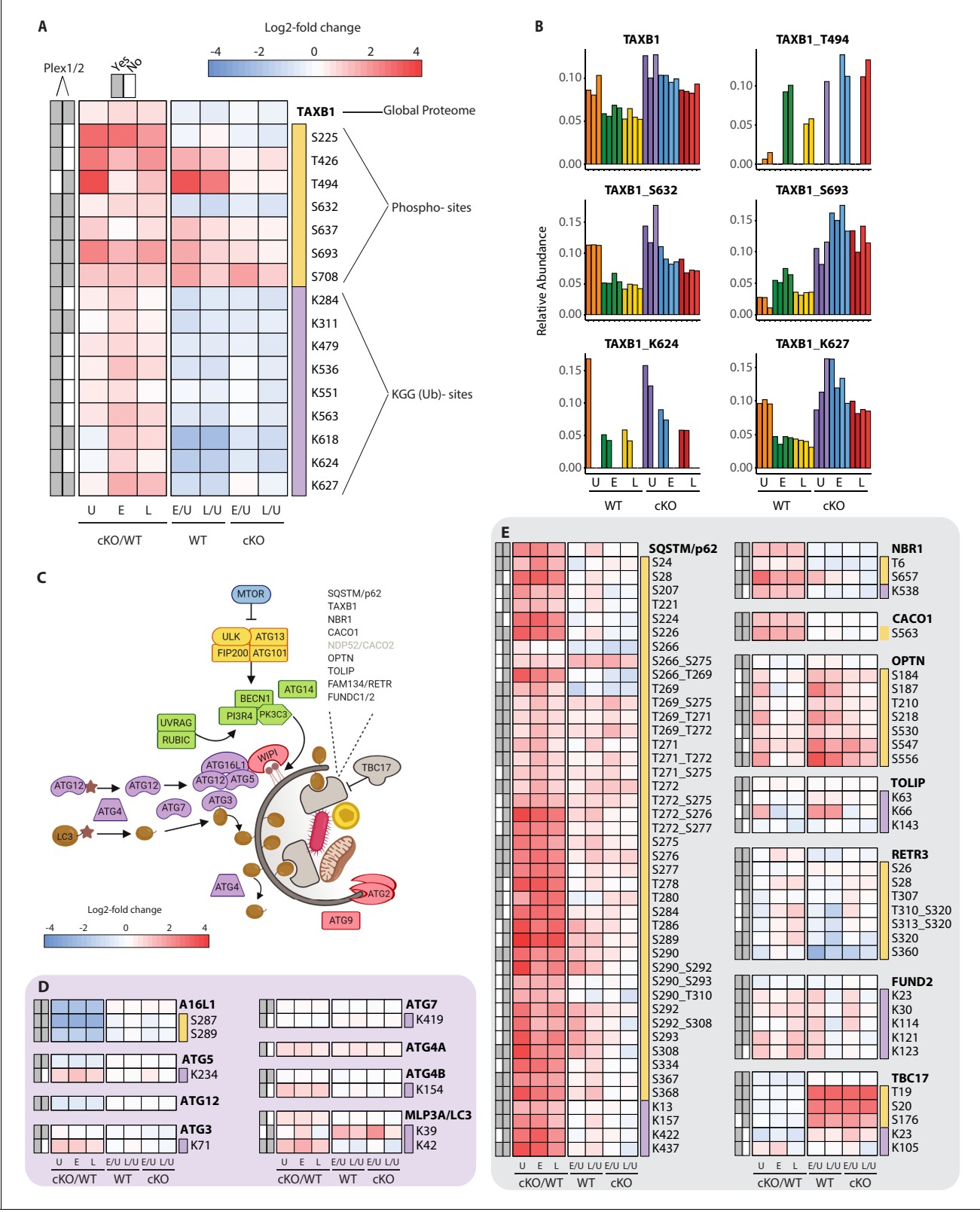

**Figure 4.** Characterization of proteomic changes in the autophagy pathway. (**A**) Heatmap representation of log2 fold changes for global proteome (unmarked), phospho-(yellow section) or KGG(Ub)-sites (purple section) measurements made for TAXB1. Data are shown for features quantified from uninfected (**U**) WT and cKO BMDMs or cells infected at early (**E**) or late (**L**) time-points with *S.flexneri*. Log2 transformed ratios are shown for contrasting genotypes (cKO/WT) at each infection timepoint (**U**, **E**, **L**) on the left and between infection timepoints (E/U and L/U) within each genotype on the right. *Figure 4 continued on next page*

*Figure 4 continued*

Gray boxes denote quantification of the feature in Plex1 and/or Plex2. Modification sites on TAXB1 denote the modified amino acid (S/T/Y/K) and residue number. (B) Bar graphs showing the relative abundance of TAXB1 global protein and representative phospho- and KGG(Ub)-sites in each of the six conditions. Note that TAXB1_K624 (Plex1) and TAXB1_T494 (Plex2) represent data collected only in a single Plex, with the relative abundance of TMT reporter ions summing up to 1.0. (C) Schematic representation of macro-autophagy and selective autophagy machinery. (D and E) Heatmap representations of E1/E2/E3-like pathway components responsible for conjugating LC3 (MLP3A) to regulate autophagosome membrane elongation (D) and selective autophagy receptors (E). The background shading for each panel corresponds to the functional color coding of proteins in the pathway schematic shown in (C). See *Supplementary file 2* for a curated list of PTMs.

The online version of this article includes the following figure supplement(s) for figure 4:

**Figure supplement 1.** Extended analysis of proteomic changes in the autophagy pathway.
**Figure supplement 2.** Characterization of proteomic changes in inflammatory signaling nodes.
**Figure supplement 3.** Analysis of proteomic changes in innate sensing and the interferon response.

We also identified notable changes across numerous components implicated in pathogen sensing such as TLRs, RLRs, NLRs and STING/cGAS (*Figure 4—figure supplement 3A and B*). Our datasets confirm several previously demonstrated PTMs that occur in response to infection, such as elevated phosphorylation of RIPK1 at S321 (*Figure 4—figure supplement 2E*), XIAP at S429 or IRF3 on multiple sites (*Figure 4—figure supplement 3D and E*). Similar effects were observed for ABIN1 (TNIP1), which showed modest changes in global protein levels, but elevated ubiquitination at multiple lysines including K360, K402, K480 at both timepoints and higher levels in cKO than WT (*Figure 4—figure supplement 2F*). Caspase-8 ubiquitination was elevated at K169 in both WT and cKO early post-infection, but was sustained through the late timepoint only in ATG16L1-deficient BMDMs (*Figure 4—figure supplement 2G*). Within the ubiquitin pathway, E3 ubiquitin ligases including HOIP (RNF31), TRAF2, and Pellino (PELI1) showed marked infection-dependent changes at the level of phosphorylation (e.g. RNF31_S445) and ubiquitination (e.g. PELI_K202 early, TRAF2_K313 late) (*Figure 4—figure supplement 2C*).

Cross-referencing all highlighted PTMs with PhosphoSitePlus revealed that ~40% were distinct from those previously identified in large-scale proteomic screens, with only ~15% of PTMs having been studied in connection to a biological function (*Supplementary files 2* and *3*). This analysis also revealed that nearly 25% of PTMs from the autophagy, innate sensing, inflammatory, and cell death signaling pathways identified in our study appear to be novel (summarized in *Table 1*).

## Elevated oxidative stress and pro-inflammatory signaling in ATG16L1-deficient macrophages contributes to accelerated bacterial killing

To obtain a global overview of the proteomics data, gene set enrichment analysis (GSEA) was performed using Hallmark gene sets to identify which signatures emerged from each of the global proteome (*Figure 5—figure supplement 1A*), phosphoproteome (*Figure 5—figure supplement 1B*), and KGG(Ub) data (*Figure 5—figure supplement 1C*). These results, particularly from the global proteome and KGG(Ub) datasets, showed a significant enrichment of pro-inflammatory gene signatures upon infection (e.g. type I interferon and TNFα signaling), but also at baseline when comparing cKO to WT cells (*Figure 5—figure supplement 1A and C*). To understand the transcriptional contribution to these proteomic signatures, RNA-Seq analysis was performed on BMDMs infected with *S.flexneri.* RNA-Seq results likewise showed a significant enrichment of pro-inflammatory gene signatures in cKO cells (e.g. type I interferon and TNFα signaling) (*Figure 5—figure supplement 2A and B*).

To explore if these pathways contribute to enhanced *S.flexneri* killing by cKO BMDMs, cells were infected in the presence of recombinant TNFα receptor II-Fc (TNFRII-Fc) to inhibit TNF signaling, or anti-IFNAR1 (α-IFNAR1) to block type I IFN signaling. In contrast to WT BMDMs (*Figure 5—figure supplement 2C and F*), inhibition of TNFα or type I interferon signaling in cKO cells resulted in a modest rescue of *S.flexneri* killing (*Figure 5—figure supplement 2D, E, G and H*), especially at later time points.

Our GSEA comparison between WT and cKO cells identified strong enrichment of the reactive oxygen species (ROS) pathway, both in the global proteome and KGG(Ub) data (*Figure 5A Figure 5—figure supplement 1A* and *Supplementary file 4* for protein set terms). This was true in the

**Table 1.** Novel post-translational modifications in specific autophagy, innate sensing, inflammatory signaling and cell death pathways revealed by TMT-MS in WT and cKO BMDMs following *S.flexneri* infection.

Please refer to heatmaps in *Figure 4*, *Figure 4—figure supplements 1*, *2* and *3* for PTM abundance changes.

**Autophagy**

| Protein name | Post-translational modification | |
|---|---|---|
| | pSTY/Phosphorylation | KGG/Ubiquitination |
| ATG5 | | K234 |
| MLP3A/LC3 | | K39 |
| TAX1BP1 | T426, T494 | K284, K311, K536, K551, K624 |
| P62/SQSTM1 | T280. S292, S308 | |
| NBR1 | T6 | |
| FUND2 | | K114, K121 |
| TBC17 | S176 | K105 |
| RBCC1/FIP200 | T642 | |
| PI3R4/VPS15 | S903, T904 | |
| RUBIC | S252, S552, S554 | |
| ATG2B | S401 | T1570 |

**Innate sensing**

| Protein name | Post-translational modification | |
|---|---|---|
| | pSTY/Phosphorylation | KGG/Ubiquitination |
| DDX58/RIG-I | | K256 |
| MAVS | Y332 | |
| CGAS | | K55 |
| TLR4 | | K692 |
| MYD88 | S136 | |
| IRAK2 | S175, T587, S615 | |
| IRAK3 | | K60, K163, K392 |
| IRAK4 | T133, S134, S175_S186 | |
| TBK1 | S509 | |
| IRF3 | T126, S130 | |
| IRF7 | S227, T277 | |
| IFIT1 | S272, S296 | K89, K117, K123, K406, K451 |
| IFIT2 | | K41, K61, K158, K291 |
| IFIT3 | S327, S333 | K246, K252, K266, K396 |
| ISG15 | K30 | |

**Inflammatory signaling, cell death**

| Protein name | Post-translational modification | |
|---|---|---|
| | pSTY/Phosphorylation | KGG/Ubiquitination |
| TNFR1B/TNFR2 | | K300 |
| M3K7/TAK1 | S331 | |
| TAB2 | S353, T376, S584 | |
| TRAF1 | | K120 |
| TRAF2 | | K194 |
| IKBz | T188 | K5, K120, K132 |
| NFKB1 | | K275 |

*Table 1 continued on next page*

| REL | S321 | |
| RNF31/HOIP | S441, S973 | K911 |
| TNAP3/A20 | S217, T567, S622, S730 | K31, K213 |
| TNIP1/ABIN1 | S601 | K288, K317, K386 |
| TNIP2/ABIN2 | T194, S196 | |
| CASP8 | S60 | K33, K274 |
| CFLAR/cFLIP | | K175, K390 |
| RIPK1 | | K429 |
| RIPK2 | S183, S381 | K369 |
| RIPK3 | S173, S177, S254, T386, T392, T398, T407 | K145, K230, K298 |

global proteome data for both uninfected and infected cells at both time-points (*Figure 5—figure supplement 1A*), as well as in the KGG(Ub) data specifically for infected cells (*Figure 5—figure supplement 1C*). This group of proteins included several factors involved in glutathione (GSH) synthesis, such as the glutamate-cysteine ligase regulatory subunit (GSH0/Glcm) and GSH synthetase (GSHB/Gss), as well as GSH regeneration, such as microsomal glutathione S-transferase (MGST1) and NAD (P)H dehydrogenase 1 (NQO1), a major enzyme that assists in reducing oxidative stress in cells (*Figure 5B*; *Hayes et al., 2020*). Additionally, several ROS converting enzymes including catalase (CATA) and peroxiredoxin 1 (PRDX1) were also elevated in cKO BMDMs at steady state. Furthermore, a subset of these redox regulators changed abundance upon *S.flexneri* infection. For example, prostaglandin dehydrogenase 1 (PGDH) displayed a time dependent decrease upon infection that was accentuated in cKO versus WT cells, consistent with its known susceptibility to ROS (*Figure 5C*; *Wang et al., 2018*). Conversely, levels of the cysteine-glutamate antiporter SLC7A11 (XCT) (*Conrad and Sato, 2012*; *Sato et al., 1999*) exhibited a significant increase in cKO BMDMs following infection (*Figure 5C*). Thus, ATG16L1 deficiency and *S.flexneri* infection might each independently elevate ROS levels, with ATG16L1 deficiency further driving a compensatory increase in the redox regulators during infection to maintain macrophage viability.

We next asked whether ATG16L1-deficient BMDMs exhibit elevated basal oxidative stress using a fluorogenic probe (CellRox green) that enables cellular ROS measurements via confocal fluorescence microscopy. Despite upregulation of numerous redox regulatory factors, CellRox probe intensity was significantly higher in cKO BMDMs (*Figure 5D and E*; see also *Figure 5—figure supplement 3A* for high content imaging). We also detected an increase in the ratio between oxidized versus reduced GSH (GSSH/GSH) in lysates of cKO BMDMs (*Figure 5—figure supplement 3B–D*). Additionally, abundance of NQO1 was significantly higher in cKO BMDMs as determined by a specific NQO1-activated fluorescent probe (*Figure 5—figure supplement 3E and F*; *Punganuru et al., 2019*). Altogether, these findings indicate that cKO BMDMs are basally exposed to higher oxidative stress, and suggest that upregulation of redox homeostasis factors is required to maintain their viability (*Tal et al., 2009*). To test this hypothesis, we treated BMDMs with Erastin, a small molecule inhibitor of XCT which increases oxidative stress by diminishing the levels of reduced GSH in cells (*Dixon et al., 2012*). Time-course studies demonstrated that BMDMs exhibited decreased viability upon prolonged XCT inhibition, and revealed a greater dependency on this pathway when *Atg16l1* is deleted (*Figure 6—figure supplement 1A*). Given the central role of autophagy in mitochondrial turnover, we analyzed mitochondrial morphology and respiration as a possible source of oxidative damage in uninfected cKO BMDMs (*Figure 5—figure supplement 3G and H*). However, mitochondrial phenotypes remained consistent between WT and cKO groups. This warrants further investigation into the underlying mechanism(s) of elevated oxidative stress in ATG16L1-deficient BMDMs and posits that altered homeostasis of other organelles may contribute to the ROS accumulation in cKO BMDMs. Interestingly, the KGG(Ub) data displayed a marked response in the peroxisome gene set in cKO cells over WT (*Figure 5—figure supplement 1C*), suggesting a possible link between this organelle and the elevated ROS signature.

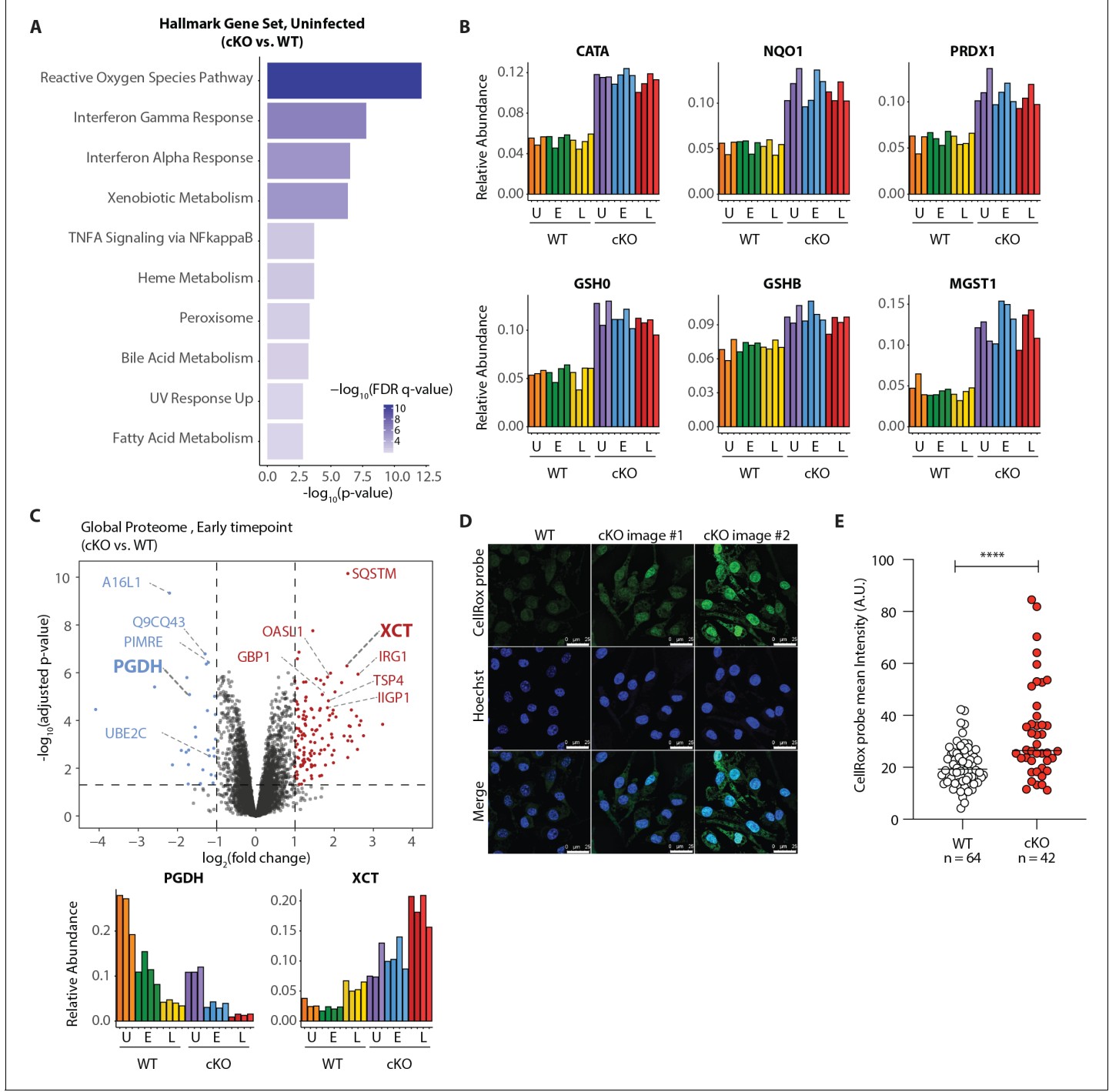

**Figure 5.** Elevated oxidative stress in ATG16L1-deficient BMDMs. (**A**) Gene set enrichment analysis (GSEA) of global proteome data showing Hallmark gene sets overrepresented in uninfected cKO over WT BMDMs. (**B**) Bar graphs show the relative abundances for selected proteins involved in redox regulation and detoxifying reactive oxygen species. (**C**) Volcano plot of global protein changes at early infection timepoint between the genotypes. Proteins enriched in cKO and WT BMDMs are highlighted in red and blue, respectively. Bar graphs showing the cumulative effects of genotype and infection on PGDH and XCT protein levels are shown below. (**D**) Representative images from experiments shown in (**E**) demonstrating CellRox probe intensity, Hoechst nuclear staining and merged images (scale bar 25 µM). (**E**) Quantification of CellRox green mean intensity in WT and cKO BMDMs. Graph shows single cell data representative of three independent experiments. Unpaired t test ****p<0.0001.

The online version of this article includes the following figure supplement(s) for figure 5:

**Figure supplement 1.** GSEA analysis of global proteome, phospho-, and KGG(Ub) datasets.

**Figure supplement 2.** Heightened pro-inflammatory signaling in cKO BMDMs contributes to enhanced *S.flexneri* killing.

*Figure 5 continued on next page*

*Figure 5 continued*

**Figure supplement 3.** Elevated oxidative stress in ATG16L1-deficient BMDMs.

Based on these observations, we hypothesized that higher levels of basal oxidative stress in cKO BMDMs may drive their enhanced ability eliminate cytosolic *S.flexneri*. To test this, we modulated oxidative stress in BMDMs using either Erastin, an oxidative stress inducer, or butylated hydroxyanisole (BHA), a ROS scavenger. BMDMs were pre-treated with Erastin or BHA prior to infection and the treatments were maintained throughout the time-course experiments (schematic in *Figure 6— figure supplement 1B*). Importantly, neither Erastin (*Figure 6—figure supplement 1C and D*) nor BHA (*Figure 6—figure supplement 1E and F*) treatments increased BMDM cell death throughout the experimental time-course. However, Erastin-treated BMDMs exhibited enhanced elimination of *S.flexneri* at early time-points following infection (*Figure 6A–C*). Conversely, BHA treatment promoted *S.flexneri* survival in BMDMs (*Figure 6D–F*). Together, these findings demonstrate a role for ATG16L1 in restraining oxidative stress, which in turn constrains macrophage microbicidal capacity.

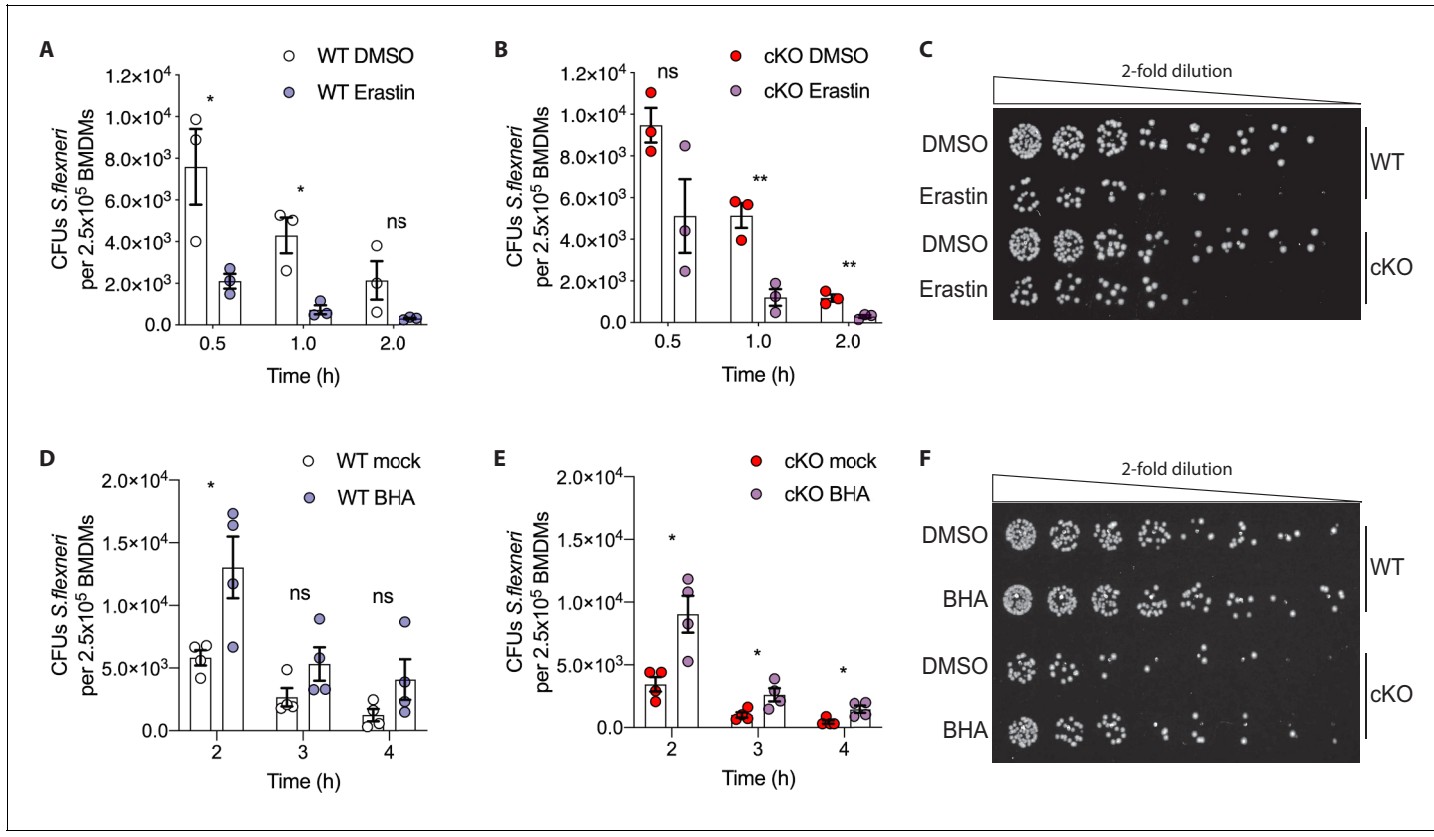

**Figure 6.** Manipulation of ROS levels modulates *S.flexneri* clearance. (A, B) Comparison of CFUs per well from three independent infection experiments in the presence of 4 µg/ml Erastin using BMDM preparations from three different *Atg16l1*-WT (A) or *Atg16l1*-cKO (B) mice. In A, 0.5 hr *p=0.04, 1 hr *p=0.01, 2 hr ns, non-significant p=0.11, multiple t-test comparison. In (B), 0.5 hr ns p=0.08, 1 hr **p=0.005, 2 hr **p=0.009, multiple t-test comparison. (C, F) Representative serial dilutions from gentamycin protection assays following *S.flexneri* M90T infection of WT or cKO BMDMs in the presence of Erastin 4 µg/ml, (C, 1 hr timepoint) or BHA 150 µM (F, 3 hr timepoint). (D, E) Comparison of CFUs per well from four independent infection experiments in the presence of 150 µM BHA using BMDM preparations from four different *Atg16l1*-WT (D) or *Atg16l1*-cKO (E) mice. In (D), 2 hr *p=0.02, 3 hr ns, non-significant p=0.13, 4 hr ns p=0.14, multiple t-test comparison. In (E), 2 hr *p=0.01, 3 hr *p=0.03, 4 hr *p=0.02, multiple t-test comparison.
The online version of this article includes the following figure supplement(s) for figure 6:

**Figure supplement 1.** Erastin, BHA sensitivity and cell death kinetics in infection experiments.

## Discussion

Emerging insights from genetic mouse models have revealed that loss of *Atg16l1* in the immune and epithelial compartments lowers the threshold for an inflammatory response (*Cadwell et al., 2010*; *Hubbard-Lucey et al., 2014*; *Lim et al., 2019*; *Matsuzawa-Ishimoto et al., 2017*). Consistently, deletion of autophagy-related genes in the innate and adaptive immune compartments have demonstrated enhanced pathogen clearance (*Marchiando et al., 2013*; *Martin et al., 2018*; *Samie et al., 2018*; *Wang et al., 2020*) as well as tumor control *in vivo* (*Cunha et al., 2018*; *DeVorkin et al., 2019*; *Lim and Murthy, 2020*). These observations prompted a re-evaluation of the pathway in antimicrobial immunity to better understand how loss of autophagy-related genes impacts cell-autonomous innate immunity against intracellular bacteria.

In this study, we show that macrophages lacking *Atg16l1* demonstrate enhanced killing of *S.flexneri*. To identify mechanisms behind this phenotype we employed TMT-based multiplexed proteomic analysis – a technology capable of near-comprehensive characterization of the global proteome (*Lapek et al., 2017*). When isobaric multiplexing methods are coupled with enrichment, it enables quantification of PTMs on thousands of individual proteins (*Rose et al., 2016*). This method is ideally suited for interrogation of a complex response such as infection of a host cell with an intracellular pathogen, where the diversity of downstream changes does not lend themselves to candidate approaches involving immunoblotting.

Our approach identifies multiple novel PTMs in components of inflammatory cytokine signaling, innate sensing and the core autophagy machinery that emerge as a consequence of *S.flexneri* infection. The comparison of early and late infection time-points shows complex dynamics that reflect PTM as well as global protein abundance. The comparison of WT versus ATG16L1-deficient BMDMs further reveals critical nodes in each of the above pathways that are under regulatory control by autophagy. The PTMs listed in *Table 1* and *Supplementary files 2* and *3* represent a sizeable fraction of the relevant post-translational changes that occur in macrophages during infection and/or upon loss of autophagy. We have provided interactive, web-accessible Spotfire Dashboards to enable user interrogation of the global proteome, phosphoproteome, and the KGG(Ub) datasets (https://info.perkinelmer.com/analytics-resource-center).

Our study reveals that basal accumulation of cellular ROS in cKO BMDMs enforces a compensatory increase in antioxidant responses exemplified by elevated protein abundances of key components of the glutathione metabolic pathway. This permits cellular viability under relatively elevated cytosolic ROS levels, which in turn suppresses *S.flexneri* expansion in BMDMs. However, overall macrophage fitness is likely compromised owing to a shift in the basal redox pathway set-point. Pharmacological depletion of GSH phenocopies genetic loss of *Atg16l1* and enhances *S.flexneri* clearance in wild-type cells, whereas the ROS scavenger BHA reverses this phenotype, demonstrating a direct role for cellular ROS in bacterial clearance upon loss of autophagy.

Mice lacking *Atg16l1* in myeloid cells also demonstrate accelerated hepatic bacterial clearance *in vivo*. However, these findings have some limitations. First, murine enteric infection by *S.flexneri* requires deletion of the NAIP-NLRC4 inflammasome in intestinal epithelial cells (recently described in *Mitchell et al., 2020*). Thus, the scope of our observations is limited by the availability of murine models and primarily reveal a role of ATG16L1 in myeloid cells during acute bacterial infection. Second, we observe a liver-specific impact of ATG16L1 deletion in the myeloid compartment because splenic *S.flexneri* colonization was comparable between genotypes, while no colonization of lung tissue was detected. However, the broad deletion of *Atg16l1* in the myeloid lineage via LysM-*Cre* raises a possibility that Kupffer cells alone may not be the sole drivers of accelerated hepatic clearance *in vivo*. Additional genetic models are required to directly test the role of autophagy in Kupffer cells, and the intestinal epithelium, as relevant sites of mucosal antimicrobial immunity.

In addition to demonstrating a role for autophagy in antimicrobial immunity via modulation of oxidative stress, our study provides the most comprehensive multiplexed proteomic analysis to date of the macrophage response to cytosolic bacterial infection. We hope this novel resource will be of broad utility to the study of myeloid signal transduction, host-pathogen interaction, and innate immunity.

# Materials and methods

## Mice

All animal experiments were performed under protocols approved by the Genentech Institutional Animal Care and Use Committee. Generation of myeloid-specific deletion of Atg16L1 was achieved by crossing *Lyz2*-Cre + mice with *Atg16L1*<sup>loxp/loxp</sup> mice (*Atg16l1*-cKO) and was described previously (*Murthy et al., 2014*). All mice were bred onto the C57BL/6N background. All *in vivo* experiments were performed using age-matched colony controls.

## Bacterial strains and culture

*Shigella flexneri* 5a strain M90T used in this study was obtained from ATCC (ATCC BAA-2402). Frozen bacterial stocks were streaked onto tryptic soy agar (TSA) plates and grown at 37°C overnight. Plates were kept at 4°C for up to 2 weeks.

## Bone-marrow-derived macrophage isolation

Femurs and tibias were collected aseptically. After removing most of the muscle and fat, the epiphyses were cut and bones were placed into PCR tubes individually hung by the hinge into a 1.5 ml Eppendorf. The bone marrow was flushed by short centrifugation at 10,000 rpm for 30 s. Red blood cells were lysed with RBC lysis buffer (Genentech) by incubating for 5 min at RT. Cells were then pelleted and resuspended in BMDM media [high glucose Dulbecco's Minimum Essential Media (DMEM) (Genentech) + 10% FBS (heat inactivated, custom manufactured for Genentech)+GlutaMAX (Gibco, 30050–061)+Pen/Strep (Gibco, 15140–122) supplemented with 50 ng/ml recombinant murine macrophage-colony stimulating factor (rmM-CSF, Genentech)] and plated in 15 cm non-TC treated dishes for 5 days (Petri dish, VWR, 25384–326). Fresh BMDM media was added on day three without removal of original media. On day 5, macrophages were gently scraped from dishes, counted and re-plated on TC-treated plates of the desired format for downstream assays in fresh BMDM media. After overnight culture in BMDM media, assays were performed on day 6 BMDMs.

## BMDM infections in 24-well plates

BMDMs isolated from control *Atg16l1*-WT or *Atg16l1*-cKO mice were plated at $2.5 \times 10^5$ cells/well in 24-well assay plates (Corning, 353047) in BMDM media. A duplicate plate was always plated for total PI-positive cell number enumeration after overnight incubation using IncuCyte ZOOM as described elsewhere. Bacterial cultures were prepared by picking a single bacterial colony from TSA plates and grown in 10 mL tryptic soy broth (TSB) in a shaking incubator overnight at 37°C. After overnight incubation, bacteria were sub-cultured in fresh 10 mL of TSB at 37°C until OD600 0.5–0.8, pelleted by centrifugation, resuspended in 1:1000 poly-L-lysine (Sigma-Aldrich, P4707) in PBS and incubated for 10 min at RT. Cell suspension was then centrifuged and the pellet washed twice with PBS and once with the infection media [high glucose DMEM (Genentech) +10% FBS (heat inactivated, custom manufactured for Genentech)+GlutaMAX (Gibco, 30050–061)]. After the final wash the bacterial pellet was resuspended in the infection media and OD600 was remeasured. To prepare multiplicity of infection (MOI) of 5 in the infection media, total PI-positive object count per well was used for accurate MOI calculations for every independent infection experiment. A cell suspension containing lysine coated bacteria were added to the wells at MOI five in a total volume of 250 μl/well and allowed to adhere by incubating for 30 min at 37°C in a $CO_2$ incubator. After 30 min, bacterial suspension was aspirated and replaced with 500 μl/well of fresh infection media supplemented with gentamicin at 50 μg/mL (Sigma-Aldrich, G1397). This was defined as the time-point T = 0 min. Assay plates were subsequently incubated at 37°C in a $CO_2$ incubator and used at the indicated time-points for CFU enumeration.

## BMDM infections in 24-well plates with compounds and antibodies

For experiments with Erastin (Sigma-Aldrich, E7781), BHA (Sigma-Aldrich, 78943), anti-IFNAR1 (Leinco Technologies, I-401), isotype control antibody clone HKSP (Leinco Technologies, I-536) or TNFRII-Fc (R and D, 9707-R2) day 5 BMDMs were plated at $2.5 \times 10^5$ cells/well in 24-well assay plates (Corning, 353047) in BMDM media and incubated overnight. A duplicate plate was also seeded and used for PI-positive object count per well enumeration to ensure accurate MOI as

described elsewhere. For experiments with Erastin, BMDMs were plated in media supplemented with Erastin at 4 µg/ml and incubated at 37°C in a $CO_2$ incubator for 18 hr before infection. For experiments with BHA, anti-IFNAR1 or TNFRII-Fc overnight media was replaced with fresh media containing 150 µM BHA, 5 µg/ml anti-IFNAR1, or 500 ng/ml TNFRII-Fc and incubated for 2 hr prior infection. The bacterial culture was prepared essentially as described elsewhere with the following modifications. After the final wash with infection media the bacterial pellet was resuspended in the infection media, OD600 was remeasured and bacterial suspension of MOI 10 was prepared. A cell suspension containing lysine coated bacteria was mixed 1:1 with infection media containing double concentrations of compounds and added to the wells in a total volume of 250 µl/well and allowed to adhere by incubating for 30 min at 37°C in a $CO_2$ incubator. After 30 min, bacterial suspension was aspirated and replaced with 500 µl/well of fresh infection media supplemented with gentamicin at 50 µg/mL (Sigma-Aldrich, G1397) and compounds as indicated.

## BMDM infections in 15 cm dishes for TMT proteomics

For large scale infections, 5 day differentiated BMDMs isolated from control *Atg16l1*-WT or *Atg16l1*-cKO mice were plated at $10 \times 10^6$ cells per 15 cm non-TC treated dish (Petri dish, VWR, 25384–326) in BMDM media. Bacterial suspension was prepared essentially as described elsewhere with the following modifications. A suspension of lysine coated bacteria in infection media were added to the dishes containing BMDMs at MOI five in a volume of 15 ml/dish and allowed to adhere by incubating for 30 min at 37°C in a $CO_2$ incubator. After 30 min, the medium was aspirated and replaced with 50 ml/dish of fresh infection media supplemented with gentamicin at 50 µg/mL (Sigma-Aldrich, G1397). This was defined as the time-point T = 0 min. Assay plates were subsequently placed at 37°C in a $CO_2$ incubator and samples collected after 30–45 min incubation ('early' infection time-point) or after 3–3.5 hr incubation ('late' infection time-point). At the indicated time-points a set of 10 dishes per genotype was used to prepare cell lysates for downstream proteomic analysis. To prepare cell lysates, infection media was first aspirated and cells washed once with PBS. Cells were then scrapped in the presence of Urea lysis buffer (20 mM HEPES pH 8.0, 9M Urea, 1 mM sodium orthovanadate, 2.5 mM sodium pyrophosphate, 1 mM β-glycerolphosphate) and cell suspension stored at −80°C until further processing (*Kirkpatrick et al., 2013*).

## In vivo *Shigella flexneri* infection

Mice were injected intravenously in the tail vein with *Shigella flexneri* (M90T) bacterial culture that was prepared by picking a single bacterial colony from TSA plates and grown in 10 mL tryptic soy broth (TSB) in a shaking incubator overnight at 37°C. After overnight incubation, bacteria were subcultured in fresh 10 mL of TSB at 37°C until OD600 0.5–0.8, pelleted by centrifugation, washed with PBS once, resuspended in PBS and OD600 was recounted. Each animal was injected with 100 µl of bacterial suspension in PBS containing $2 \times 10^6$ Colony Forming Units (CFUs) *S.flexneri* strain M90T. Mice were euthanized six or 24 hr post infection to harvest spleen, liver, and lung for CFUs enumeration.

## Colony-forming units (CFUs) assays

To determine CFUs in infected BMDMs, infection media was aspirated, cells were washed once with PBS and lysed by adding 250 µl/well of 0.1% Igepal CA-630 (Sigma-Aldrich, I8896) in PBS, incubated for 5 min, resuspended and an aliquot of 200 µl was transferred to 96-well U-bottom plate (Costar, 3799) for making two-step serial dilutions in 0.1% Igepal CA-630 in PBS. Subsequently, 5 µl of each serial dilution was plated on TSA plates in triplicates, allowed to evaporate at RT after which the plate was placed in a 37°C incubator overnight. After overnight incubation, colonies from individual dilutions were counted and used for determining CFUs per well. To determine CFUs in the liver, spleen, and lung, mice were euthanized at the indicated time-points after infection and the organs surgically removed and placed in PBS on ice. Livers were processed in 5 ml of 0.1% Igepal CA-630 (Sigma-Aldrich, I8896) in PBS using the gentleMACS C Tubes (Miltenyi Biotec, 130-096-334) in combination with the gentleMACS Octo Dissociator (Miltenyi Biotec, 130-095-937) for the automated dissociation of tissues using standard tissue dissociation programs (program sequence: m_liver_01_02; m_liver_02_02, m_liver_01_02). Tissue suspensions were filtered through 100 µM filters (CellTreat, 229485) and remaining liver tissue was additionally homogenized using the rubber

seal of the 5 ml syringe plunger. For the spleen and lung, organs were processed in 2 ml of 0.1% Igepal and dissociated using the rubber seal of the 5 ml syringe plunger before filtering through 70 μM filters. The resultant tissue suspension was used for generating serial dilutions and plated on TSA plates for CFUs enumeration as described elsewhere.

## IncuCyte assays

For IncuCyte assays, BMDMs were plated at $2 \times 10^4$ cells/well in flat-bottom 96-well (Corning, 353072) or at $2.5 \times 10^5$ cells/well in 24-well (Corning, 353047) assay plates. After overnight incubation at 37°C in a $CO_2$ incubator, cells were used for infection experiments or treatments with compounds or growth factors as indicated. BMDM viability over time was assessed by supplementing assay media [(high glucose DMEM (Genentech) +10% FBS (heat inactivated, custom manufactured for Genentech)+GlutaMAX (Gibco, 30050–061)+Pen/Strep (Gibco, 15140–122))] with propidium iodide (PI) dye for live-cell imaging at 1:1000 (Invitrogen, P3566), and then measuring PI-positive cells per $mm^2$ using live cell imaging with IncuCyte ZOOM (IncuCyte systems, Essen Biosciences) in a time-course experiment. Percent cell death was calculated by dividing PI-positive cells per $mm^2$ with total plated cells per $mm^2$. Total plated cells were enumerated from a duplicate plate seeded at the same time as the assay plates. After overnight incubation, media in the duplicate plate was exchanged to assay media containing 0.06 % NP-40 supplemented with 1:1000 PI, and imaged at a single time-point using IncuCyte ZOOM after 10 min incubation.

## GSH assays

BMDMs were established as described and $5 \times 10^6$ of BMDMs were pelleted by centrifugation, the pellet was lysed in mammalian lysis buffer (Abcam, ab179835), incubated 10 min at RT and centrifuged at top speed at 4°C 15 min. Supernatant was transferred to a fresh tube and used for deproteinization following manufacturer's instructions (Abcam, ab204708). The resultant supernatant was used for determining reduced GSH, total GSH and oxidized GSSG was calculated as per manufacturer's instructions (Abcam, ab138881).

## Flow cytometry analysis

The NQO1 activity probe was generated in-house as described in *Punganuru et al., 2019*. The endogenous NQO1 activity of WT and cKO BMDMs was determined by flow cytometry after exposure to the probe. Cells were incubated with 10 μM probe for 60 min, harvested by non-enzymatic dissociation (Gibco, 13150–016) and washed with PBS. About $1 \times 10^4$ cells were analyzed by flow cytometry (λex = 488 nm). Mean fluorescence intensity (MFI) readout from each independent experiment was used for the graph.

## RNA-Seq

BMDMs were established from three pairs of control *Atg16l1*-WT or *Atg16l1*-cKO mice and infected with *S.flexneri* strain M90T as described elsewhere. Samples were collected after 45 min incubation ('early' infection time-point), 3.5 hr incubation ('late' infection time-point) or from uninfected samples. RNA was isolated according to manufacturer's protocol (Qiagen, 74104) and RNA-sequencing data were analyzed using HTSeqGenie (*Pau and Reeder, 2021*) in BioConductor (*Huber et al., 2015*) as follows: first, reads with low nucleotide qualities (70% of bases with quality <23) or rRNA and adapter contamination were removed. The reads that passed were then aligned to the mouse reference genome GRCm38.p5 using GSNAP (*Wu and Nacu, 2010*) version '2013-10-10-v2' with alignment parameters: '-M 2 n 10 -B 2 -i 1 N 1 w 200000 -E 1 --pairmax-rna=200000'. Alignments of the reads that were reported by GSNAP as 'uniquely mapping' were used for subsequent analysis. Gene expression levels were quantified as Reads Per Kilobase of exon model per Million mapped reads normalized by size factor (nRPKM), defined as number of reads aligning to a gene in a sample / (total number of uniquely mapped reads for that sample x gene length x size factor). Differential expression and gene set enrichment analysis were performed with voom +limma [PMID 27441086] and bioconductor package *fgsea* (Fast Gene Set Enrichment Analysis) (*Korotkevich et al., 2021*), respectively.

## Fluorescence microscopy

BMDMs grown on 96-well plates (Greiner Bio, 655090) were treated with 10 µM CellRox Green reagent for 30 min according to manufacturer's protocol (Thermo Fisher Scientific, C10444), then fixed in 4% paraformaldehyde (PFA) solution in PBS (ChemCruz, SC281692) for 15 min at RT. Nuclei were stained with NucBlue Live ReadyProbes Reagent (Thermo Fisher Scientific, R37605) for 10 min in PBS. 3D confocal images corresponding to 12-µm-thick z-stacks of four stitched fields of views were collected on a Nikon A1R scanning confocal microscope using a Plan Apo NA 0.75 lens and x20 magnification. FITC and Hoechst 33342 signals were respectively imaged with the 488 nm and 405 nm laser lines. For each Z stack, images were combined into one focused image using Nikon Elements Extended Depth of focus (EDF) module that picks the focused regions from each frame and merges them together into a single focused image. The focused EDF images from different conditions were then analyzed with Bitplane Imaris software (version 9.2.0) using the cell segmentation module and intensity quantification. To specifically determine the cytoplasmic CellRox Green reagent intensity, the region corresponding to the Hoechst staining was excluded and FITC channel threshold was applied across all samples per given experiment. Mean cytosolic CellRox Green assay signal was then quantified per each individual cell and presented in the graphs.

## Transmission electron microscopy

Samples were fixed in modified Karnovsky's fixative (2% paraformaldehyde and 2.5% glutaraldehyde in 0.1M sodium cacodylate buffer, ph7.2) and then post-fixed in freshly prepared 1% aqueous potassium ferrocyanide- osmium tetroxide (EM Sciences, Hatfield, PA), for 2 hr followed by overnight incubation in 0.5% Uranyl acetate at 4°C. The samples were then dehydrated through ascending series of ethanol (50%, 70%, 90%, 100%) followed by propylene oxide (each step was for 15 min) and embedded in Eponate 12 (Ted Pella, Redding, CA). Ultrathin sections (80 nm) were cut with an Ultracut microtome (Leica), stained with 0.2% lead citrate and examined in a JEOL JEM-1400 transmission electron microscope (TEM) at 80kV. Digital images were captured with a GATAN Ultrascan 1000 CCD camera.

## Tandem mass tag proteomics

### Protein precipitation

Protein concentration in the lysates were quantified using the Pierce micro-BCA assay (ThermoFisher Scientific, Waltham, MA). All protein from the cell lysates were precipitated with a combination of methanol/chloroform/water (*Wessel and Flügge, 1984*). In brief, 1X volume of lysate was mixed with 4X volume of methanol followed by 2X volume of chloroform and 3X volume of water. The protein pellets were washed a total of three times with 5X volume of methanol. The protein pellets were air dried and resuspended in 8M urea, 100 mM EPPS pH 7.0, 5 mM DTT. Proteins were alkylated with 15 mM N-ethylmaleimide (Sigma).

### LysC/trypsin digestion

The protein in 8M urea was diluted to 4M with 100 mM EPPS, pH 8.0. 15 mg of protein/sample was digested at 25°C for 12 hr with lysyl endopeptidase (LysC, Wako Chemicals USA) at a 1:25; protein: protease ratio. Following LysC digestion the peptides in 4M urea were diluted to 1M urea with 100 mM EPPS, pH 8.0. The LysC peptides were digested with trypsin at 37 °C for 8 hr (Promega) at a 1:50; protein:protease ratio.

### Ubiquitin remnant peptide enrichment (KGG peptides)

Prior to KGG peptide enrichment, the tryptic peptides were acidified to 2% formic acid and desalted with 1 g tC18 Sep-Pak cartridges (Waters). The desalted peptides were dried by vacuum. KGG peptide enrichment was performed with the PTMScan ubiquitin remnant motif kit (Cell Signaling Technologies, Kit#5562) as per the manufacturers protocol. KGG peptides eluted from the antibodies were dried by vacuum. The flow through peptides from the KGG enrichment were saved for phosphopeptide and total protein analysis.

## TMT labeling of KGG peptides

Peptides were resuspended in 200 mM EPPS, pH 8.0. 10 µL of TMT reagent at 20 µg/uL (Thermo-Fisher) was added to each sample. Peptides were incubated with TMT reagent for 3 hr at 25℃. TMT-labeled peptides were quenched with hydroxylamine (0.5% final) and acidified with trifluoro-acetic acid (2% final). The samples were combined, desalted with 50 mg tC18 Sep-Paks, and dried by vacuum.

## Ubiquitin remnant peptide fractionation

TMT-labeled KGG peptides were fractionated using the high pH reversed-phase peptide fraction-ation kit (ThermoFisher). The dried KGG peptides were resuspended in 0.1% trifluoroacetic acid and fractionated according to the manufacturers protocol into six fractions (17.5%, 20%, 22.5%, 25%, 30%, and 70% acetonitrile +0.1% triethylamine). The KGG peptide fractions were dried by vacuum, desalted with StageTips packed with Empore C18 material (3M, Maplewood, MN.), and dried again by vacuum. KGG peptides were reconstituted in 5% formic acid +5% acetonitrile for LC-MS3 analysis.

## TMT labeling of KGG flow through peptides

The flow through peptides from the KGG enrichment were labeled with TMT prior to phosphopep-tide enrichment. The flow through peptides were resuspended in 1X IAP buffer from the ubiquitin remnant kit (from prior step). The pH of the resuspended peptides was adjusted by adding 1M EPPS, pH 8.0 in a 3:1 ratio (peptide volume:1M EPPS volume; 250 mM EPPS final). 2.1 mg of peptide from each sample was labeled with 2.4 mg of TMT reagent resuspended in 60 µL, 100% acetonitrile. The peptides were incubated with TMT reagent for 3 hr at 25℃. TMT-labeled peptides were quenched with hydroxylamine (0.5% final) and acidified with trifluoroacetic acid (2% final). The sam-ples were combined, desalted with 1 g tC18 Sep-Paks, dried by vacuum.

## Phosphoserine, -threonine, -tyrosine enrichment and fractionation

Phosphotyrosine (pY) peptides were enriched using the Cell Signaling Technologies pY-1000 anti-body kit as per the manufacturers protocol (Cell Signaling Technologies, Kit#8803). The flow through from the pY enrichment was desalted on a 1 g tC18 Sep-Pak cartridge (Waters Corporation, Milford, MA) and dried by centrifugal evaporation and saved for phosphoserine and phosphothreonine (pST) analysis. pST phosphopeptides were enriched using the Pierce Fe-NTA phospho-enrichment kit (ThermoFisher). In brief, peptides were bound and washed as per manufacturers protocol. Phospho-peptides were eluted from the Fe-NTA resin with 50 mM $HK_2PO_4$ pH 10.5. Labeled phosphopepti-des were subjected to orthogonal basic-pH reverse phase fractionation on a $3 \times 100$ mm column packed with 1.9 µm Poroshell C18 material (Agilent, Santa Clara, CA), utilizing a 45 min linear gradi-ent from 8% buffer A (5% acetonitrile in 10 mM ammonium bicarbonate, pH 8) to 30% buffer B (ace-tonitrile in 10 mM ammonium bicarbonate, pH 8) at a flow rate of 0.4 ml/min. Ninety-six fractions were consolidated into 18 samples, acidified with formic acid and vacuum dried. The samples were resuspended in 0.1% trifluoroacetic acid, desalted on StageTips and vacuum dried. Peptides were reconstituted in 5% formic acid +5% acetonitrile for LC-MS3 analysis. The flow-through peptides from the pST enrichment were saved for total protein analysis.

## Peptide fractionation for total protein analysis

The flow-through from the pST enrichment was dried by centrifugal evaporation. The dried peptides were resuspended in 0.1% TFA. Approximately 250 µg of peptide mix was subjected to orthogonal basic-pH reverse phase fractionation on a $3 \times 100$ mm column packed with 1.9 µm Poroshell C18 material (Agilent, Santa Clara, CA), utilizing a 45 min linear gradient from 8% buffer A (5% acetoni-trile in 10 mM ammonium bicarbonate, pH 8) to 35% buffer B (acetonitrile in 10 mM ammonium bicarbonate, pH 8) at a flow rate of 0.4 ml/min. Ninety-six fractions were consolidated into 12 sam-ples, acidified with formic acid and vacuum dried. The samples were resuspended in 5% formic acid, desalted on StageTips and vacuum dried. Peptides were reconstituted in 5% formic acid +5% aceto-nitrile for LC-MS3 analysis.

## Mass spectrometry analysis

All mass spectra were acquired on an Orbitrap Fusion Lumos coupled to an EASY nanoLC-1000 (or nanoLC-1200) (ThermoFisher) liquid chromatography system. Approximately 2 μg of peptides were loaded on a 75 μm capillary column packed in-house with Sepax GP-C18 resin (1.8 μm, 150 Å, Sepax Technologies) to a final length of 35 cm. Peptides for total protein analysis were separated using a 180 min linear gradient from 8% to 23% acetonitrile in 0.1% formic acid. The mass spectrometer was operated in a data-dependent mode. The scan sequence began with FTMS1 spectra (resolution = 120,000; mass range of 350–1400 *m/z*; max injection time of 50 ms; AGC target of 1e6; dynamic exclusion for 60 s with a ± 10 ppm window). The ten most intense precursor ions were selected for ITMS2 analysis via collisional-induced dissociation (CID) in the ion trap (normalized collision energy (NCE) = 35; max injection time = 100 ms; isolation window of 0.7 Da; AGC target of 2e4). Following ITMS2 acquisition, a synchronous-precursor-selection (SPS) MS3 spectrum was acquired by selecting and isolating up to 10 MS2 product ions for additional fragmentation via high energy collisional-induced dissociation (HCD) with analysis in the Orbitrap (NCE = 55; resolution = 50,000; max injection time = 110 ms; AGC target of 1.5e5; isolation window at 1.2 Da for +2 *m/z*, 1.0 Da for +3 *m/z* or 0.8 Da for +4 to+6 *m/z*). pY peptides were separated using a 180 min linear gradient from 7% to 26% acetonitrile in 0.1% formic acid. The mass spectrometer was operated in a data dependent mode. The scan sequence began with FTMS1 spectra (resolution = 120,000; mass range of 350–1400 *m/z*; max injection time of 50 ms; AGC target of 1e6; dynamic exclusion for 75 s with a ± 10 ppm window). The ten most intense precursor ions were selected for FTMS2 analysis via collisional-induced dissociation (CID) in the ion trap (normalized collision energy (NCE) = 35; max injection time = 150 ms; isolation window of 0.7 Da; AGC target of 3e4; *m/z* = 2–6; Orbitrap resolution = 15 k). Following FTMS2 acquisition, a synchronous-precursor-selection (SPS) MS3 method was enabled to select five MS2 product ions for high energy collisional-induced dissociation (HCD) with analysis in the Orbitrap (NCE = 55; resolution = 50,000; max injection time = 300 ms); AGC target of 1e5; isolation window at 1.2 Da. pST peptides were separated using a 120 min linear gradient from 6% to 26% acetonitrile in 0.1% formic acid. The mass spectrometer was operated in a data-dependent mode. The scan sequence began with FTMS1 spectra (resolution = 120,000; mass range of 350–1400 *m/z*; max injection time of 50 ms; AGC target of 1e6; dynamic exclusion for 60 s with a ± 10 ppm window). The ten most intense precursor ions were selected for ITMS2 analysis via collisional-induced dissociation (CID) in the ion trap (normalized collision energy (NCE) = 35; max injection time = 200 ms; isolation window of 0.7 Da; AGC target of 2e4). Following MS2 acquisition, a synchronous-precursor-selection (SPS) MS3 method was enabled to select five MS2 product ions for high-energy collisional-induced dissociation (HCD) with analysis in the Orbitrap (NCE = 55; resolution = 50,000; max injection time = 300 ms; AGC target of 1e5; isolation window at 1.2 Da for +2 *m/z*, 1.0 Da for +3 *m/z* or 0.8 Da for +4 to+6 *m/z*).

KGG peptides were separated using a 180 min linear gradient from 7% to 24% acetonitrile in 0.1% formic acid. The mass spectrometer was operated in a data dependent mode. The scan sequence began with FTMS1 spectra (resolution = 120,000; mass range of 350–1400 *m/z*; max injection time of 50 ms; AGC target of 1e6; dynamic exclusion for 75 s with a ± 10 ppm window). The 10 most intense precursor ions were selected for FTMS2 analysis via collisional-induced dissociation (CID) in the ion trap (normalized collision energy (NCE) = 35; max injection time = 100 ms; isolation window of 0.7 Da; AGC target of 5e4; *m/z* 3–6, Orbitrap resolution set to 15 k). Following MS2 acquisition, a synchronous-precursor-selection (SPS) MS3 method was enabled to select 10 MS2 product ions for high-energy collisional-induced dissociation (HCD) with analysis in the Orbitrap (NCE = 55; resolution = 50,000; max injection time = 500 ms; AGC target of 1e5; isolation window at 1.0 Da for +3 *m/z* or 0.8 Da for +4 to+6 *m/z*).

MS/MS spectra for the global proteome, serine/threonine phosphorylated, tyrosine phosphorylated, and ubiquitylated data sets were searched using the Mascot search algorithm (Matrix Sciences) against a concatenated target−decoy database comprised of the UniProt mouse and *Shigella flexneri* protein sequences (version 2017_08), known contaminants and the reversed versions of each sequence. For all datasets, a 50 ppm precursor ion mass tolerance was selected with tryptic specificity up to two missed cleavages. For the global proteome and serine/threonine phosphorylated datasets a 0.8 Da fragment ion tolerance was selected. While for the tyrosine phosphorylated and KGG (ubiquitin) datasets a 0.02 Da fragment ion tolerance was selected. The global proteome and

phosphorylated datasets used a fixed modification of N-ethylmaleimide on cysteine residues (+125.0477) as well as TMT 11-plex on Lysine and the peptide N-term (+229.1629). The ubiquity-lated data set used a fixed modification of N-ethylmaleimide on cysteine residues (+125.0477) as well as TMT 11-plex on the peptide N-term (+229.1629). For variable modifications the global prote-ome dataset used methionine oxidation (+15.9949) as well as TMT 11-plex on tyrosine (+229.1629). The phosphorylated dataset used the same variable modifications as the global proteome dataset plus phosphorylation on serine, threonine, and tyrosine (+79.9663). Finally, the KGG(Ub) dataset used methionine oxidation (+15.9949), TMT 11 plex on tyrosine and lysine (+229.1629), as well as TMT 11 Plex +ubiquitylation on lysine (343.2059). PSMs were filtered to a 1% peptide FDR at the run level using linear discriminant analysis (LDA) (*Kirkpatrick et al., 2013*). PSM data within each plex and dataset (global proteome, phosphorylation, and ubiquitylation) was aggregated and these results were subsequently filtered to 2% protein FDR. For PSMs passing the peptide and protein FDR filters within the phosphorylated and ubiquitylated datasets, phosphorylation and ubiquitylation site localization was assessed using a modified version of the AScore algorithm (*Beausoleil et al., 2006*) and reassigned accordingly. Finally, reporter ion intensity values were determined for each dataset and plex using the Mojave algorithm (*Zhuang et al., 2013*) with an isolation width of 0.7.

## Quantification and statistical analysis of global proteomics and phosphoproteomic data

Quantification and statistical testing of global proteomics data were performed by MSstatsTMT v1.8.0, an open-source R/Bioconductor package (*Huang et al., 2020*; *Tsai et al., 2020*). MSstatsTMT was used to create quantification reports and statistical testing reports using the Peptide Spectrum Matches (PSM) as described above. First, PSMs were filtered out if they were (1) from decoy proteins; (2) from peptides with length less than 7; (3) with isolation specificity less than 70%; (4) with reporter ion intensity less than 2eight noise estimate; (5) from peptides shared by more than one protein; (6) with summed reporter ion intensity (across all eleven chan-nels) lower than 30,000; (7) with missing values in more than nine channels. In the case of redun-dant PSMs (i.e. multiple PSMs in one MS run corresponding to the same peptide ion), only the single PSM with the least missing values or highest isolation specificity or highest maximal reporter ion intensity was retained for subsequent analysis. Multiple fractions from the same TMT mixture were combined in MSstatsTMT. In particular, if the same peptide ion was identified in multiple fractions, only the single fraction with the highest mean or maximal reporter ion intensity was kept. Next, MSstatsTMT generated a normalized quantification report across all the samples at the protein level from the processed PSM report. Global median normalization, which equal-ized the median of the reporter ion intensities across all the channels and TMT mixtures, was car-ried out to reduce the systematic bias between channels. The normalized reporter ion intensities of all the peptide ions mapped to a protein were summarized into a single protein level intensity in each channel and TMT mixture. For each protein, additional local normalization on the summa-ries was performed to reduce the systematic bias between different TMT mixtures. For the local normalization, we created an artifact reference channel by averaging over all the channels except 131C for each protein and TMT mixture. The channel 131C was removed in order to make each mixture have the same number of samples from each condition. The normalized quantification report at the protein level is available in *Supplementary file 5*. As a final step, the differential abundance analysis between conditions was performed in MSstatsTMT based on a linear mixed-effects model per protein. The inference procedure was adjusted by applying an empirical Bayes shrinkage. To test two-sided null hypothesis of no changes in abundance, the model-based test statistics were compared to the Student t-test distribution with the degrees of freedom appropri-ate for each protein or each PTM site. The resulting p values were adjusted to control the FDR with the method by Benjamini-Hochberg. The table with the statistical testing results for all the proteins is available as in *Supplementary file 6*. Quantification and statistical testing for phos-pho- and KGG(Ub)-sites data were performed by the same procedure as for global proteomics data with some modifications. First, PSMs from non-modified peptides were filtered out from the PSM report and the remaining preprocessing analyses were the same as above. Second, custom PTM site identifiers were created for each PSM by identifying the modified residue index in the reference proteome that was used to search the MS/MS spectra. Finally, all steps for

quantification and differential abundance analysis were performed at the PTM site level, rather than the protein level (*Supplementary files 7*, *8*, *9* and *10*). The relative abundance of TMT reporter ion abundances in bar graphs throughout the paper stems from MSstatsTMT modeling and sums up to 1.0 for each Plex. Thus, the sum of all signal shown sums to 1.0 or 2.0 depending on whether the feature was quantified in one or both plexes. For the consolidated heatmaps showing proteome level changes immediately adjacent to any identified PTMs, the ComplexHeatmap R package was used.

## Overview heatmaps/clustering

For the overview heatmaps showing PTM and global proteome datasets side by side, clustering was performed as follows. First, protein quantification results from MSstatsTMT for the PTM and global proteome datasets were merged with the phospho-proteome and KGG datasets, respectively. For each of the two combined datasets, the pheatmap R package was used to cluster the protein model results into 16 row wise clusters using the clustering method 'ward.D'. The columns of the dataset were kept static and not clustered.

## Gene set enrichment analysis

Gene set enrichment analysis was performed using MsigDB (*Liberzon et al., 2015*; *Subramanian et al., 2005*). Global proteome data were filtered to include features with an absolute value log2fc values of greater than one as well as adjusted p values of less than 0.05. Subsequently the data were filtered to require that every protein must be found in both multiplexed experiments. UniProt identifiers were transformed to gene symbols and fed into GSEA for an enrichment analysis against MsigDB's hallmark gene sets. Gene set enrichment results were filtered to 5% FDR.

## Statistical analysis

Pairwise statistical analyses were performed using an unpaired t-test using two-stage step-up method of Benjamini, Krieger and Yekutieli and false discovery rate of 1% to determine if the values in two sets of data differ. Multiple-comparison corrections were made using the Sidak method with family-wise significance and confidence level of 0.05. Analysis of *in vivo* infection data was done using unpaired two-tailed t-test after outliers were removed using ROUT method (Q = 1%). Analysis of kinetic (time) with Erastin was performed using two-way ANOVA followed by multiple comparison testing. Line graphs and associated data points represent means of data; error bars represent standard deviation from mean. GraphPad Prism eight software was used for data analysis and representation. p-Values: *<0.05, **<0.01, ***<0.001, ****<0.0001.

## The software availability statement

Raw files were converted to mzXML using ReadW (v 4.3.1) available through https://sourceforge. net/projects/sashimi/files/ReAdW%20%28Xcalibur%20converter%29/. Spectra were searched using Mascot (v 2.4.1) licensed from Matrix Sciences. Search results were filtered using the LDA function in the MASS Package in R as described in Huttlin et al. Cell 143, 1147–1189 (2010). Mojave is an in-house tool developed to report TMT reporter ion intensity values and is available upon request. MSstatsTMT (v 1.8.0) is a freely available open-source R/Bioconductor package to detect differentially abundant proteins in TMT experiments. It can be installed through https://www.bioconductor. org/packages/release/bioc/html/MSstatsTMT.html. Gene set enrichment was performed using the GSEA/MSigDB web portal https://www.gsea-msigdb.org/gsea/msigdb/annotate.jsp. Heatmaps were generated using the pheatmap (v1.0.12) (https://cran.r-project.org/web/packages/pheatmap/index.html) or ComplexHeatmap (v 2.4.2) (https://bioconductor.org/packages/release/bioc/html/ComplexHeatmap.html) R packages.

## Acknowledgements

This work was funded in part by a fellowship awarded to TM by the AXA Research fund (16-AXA-PDOC-078) and the European Research Council (ERC) under the European Union's Horizon 2020 research and innovation program to ID (grant agreement No. 742720). We thank the Genentech Visiting Scientist Program and Research Innovation Fund for supporting this work. We also thank

Avinashnarayan Venkatanarayan, Beatrice Breart and the laboratory of Eric Brown at Genentech for technical assistance.

## Additional information

### Competing interests

Ivan Dikic: Reviewing editor, *eLife*. Timurs Maculins, Trent Hinkle, Patrick Chang, Cecile Chalouni, Youngsu Kwon, Junghyun Lim, Anand Kumar Katakam, Mike Reichelt, Yasin Senbabaoglu, Brent Mckenzie: is an employee of Genentech, Inc and a shareholder in Roche. Erik Verschueren: is a current employee at Galapagos. Meena Choi: was employed at Northeastern Univeristy during the preparation of the manuscript and is currently an employee of Genentech Inc and a shareholder in Roche. Shilpa Rao: was employed at Genentech, Inc during the preparation of the manuscript. Ryan C Kunz, Brian K Erickson: is an employee of IQ Proteomics LLC. Donald S Kirkpatrick, Aditya Murthy: is an employee of Interline Therapeutics. The other authors declare that no competing interests exist.

### Funding

| Funder | Grant reference number | Author |
|---|---|---|
| AXA Research Fund | 16-AXA-PDOC-078 | Timurs Maculins |
| Genentech | | Timurs Maculins |
| H2020 European Research Council | 742720 | Ivan Dikic |

The funders had no role in study design, data collection and interpretation, or the decision to submit the work for publication.

### Author contributions

Timurs Maculins, Conceptualization, Data curation, Formal analysis, Validation, Investigation, Visualization, Methodology, Writing - original draft, Writing - review and editing; Erik Verschueren, Meena Choi, Data curation, Software, Formal analysis, Visualization, Methodology, Writing - review and editing, TMT data acquisition, analysis and representation; Trent Hinkle, Data curation, Formal analysis, Visualization, Methodology, Writing - review and editing; Patrick Chang, Cecile Chalouni, Investigation, Methodology, Immunofluorescence microscopy data acquisition and quantification; Shilpa Rao, Formal analysis, Visualization, Added during revision: Performed analysis and visualization of new RNA-Seq data presented in the revision (Figure 5-figure supplement 2A, B); Youngsu Kwon, Investigation, Methodology, Added during revision: Performed additional in vivo studies presented in the revision (Figure 1D, E; Figure 1-figure supplement 1B; Figure R1); Junghyun Lim, Investigation, Methodology, Assisted in design and execution of large-scale proteomic experiments; Anand Kumar Katakam, Data curation, Formal analysis, Investigation, Methodology, Electron microscopy data acquisition, analysis and representation; Ryan C Kunz, Brian K Erickson, Formal analysis, Methodology, TMT data acquisition and initial data analysis; Ting Huang, Software, Formal analysis, TMT data acquisition, analysis and representation; Tsung-Heng Tsai, Olga Vitek, Software, Formal analysis, TMT data acquisition and initial data analysis; Mike Reichelt, Supervision, Investigation, Methodology; Yasin Senbabaoglu, Formal analysis, Supervision, Methodology, Added during revision: Designed RNA-Seq study, supervised analysis and visualization of new RNA-Seq data presented in the revision (Figure 5-figure supplement 2A, B); Brent Mckenzie, Resources, Supervision, Project administration, Added during revision: Guided and supervised in vivo studies presented in the revision (Figure 1D, E; Figure 1-figure supplement 1B; Figure R1); John R Rohde, Resources, Investigation, Writing - review and editing, Generation of S.flexneri strains. Guidance for in vitro BMDM infection experiments; Ivan Dikic, Conceptualization, Supervision, Funding acquisition, Writing - original draft, Writing - review and editing; Donald S Kirkpatrick, Conceptualization, Resources, Supervision, Visualization, Writing - original draft, Writing - review and editing, Designed the conceptual framework of the study and experiments. TMT data acquisition, analysis and representation; Aditya

Murthy, Conceptualization, Resources, Supervision, Funding acquisition, Investigation, Visualization, Methodology, Writing - original draft, Project administration, Writing - review and editing

### Author ORCIDs
Timurs Maculins  https://orcid.org/0000-0002-9333-301X
Meena Choi  https://orcid.org/0000-0002-6025-5035
Ting Huang  http://orcid.org/0000-0003-1041-2167
Tsung-Heng Tsai  http://orcid.org/0000-0001-5211-2985
Ivan Dikic  https://orcid.org/0000-0001-8156-9511
Aditya Murthy  https://orcid.org/0000-0002-6130-9568

### Ethics
Animal experimentation: All animal experiments were performed under protocols approved by the Genentech Institutional Animal Care and Use Committee. Generation of myeloid-specific deletion of Atg16L1 was achieved by crossing LysM-Cre+ mice with Atg16L1loxp/loxp mice and was described previously (Murthy et al., 2014). All mice were bred onto the C57BL/6N background. All in vivo experiments were performed using age-matched colony controls. Protocol ID 17-2842.

### Decision letter and Author response
Decision letter https://doi.org/10.7554/eLife.62320.sa1
Author response https://doi.org/10.7554/eLife.62320.sa2

## Additional files
### Supplementary files
• Supplementary file 1. Composition PTM-site and global protein clusters displayed in *Figure 2F and G* and *Figure 2—figure supplement 1B,C*.

• Supplementary file 2. Curated list of PTMs described in *Figure 4* and *Figure 4—figure supplement 1* with associated references.

• Supplementary file 3. Curated list of PTMs described in *Figure 4—figure supplement 2* and *Figure 4—figure supplement 3* with associated references.

• Supplementary file 4. Gene Set Enrichment Analysis (GSEA) performed to identify cellular processes overrepresented in ATG16L1 deficient BMDMs in *Figure 5A*.

• Supplementary file 5. MSstatsTMT normalized quantification report for global proteins data.

• Supplementary file 6. MSstatsTMT statistical testing results for global proteins data.

• Supplementary file 7. MSstatsTMT normalized quantification report for phospho-sites data.

• Supplementary file 8. MSstatsTMT statistical testing results for phospho-sites data.

• Supplementary file 9. MSstatsTMT normalized quantification report for KGG(Ub)-sites data.

• Supplementary file 10. MSstatsTMT statistical testing results for KGG(Ub)-sites data.

• Transparent reporting form

### Data availability
Mass spectrometry raw files have been uploaded to the UCSD MassIVE repository and are available: (https://massive.ucsd.edu/ProteoSAFe/dataset.jsp?accession=MSV000085565; Password = shigella).

The following datasets were generated:

| Author(s) | Year | Dataset title | Dataset URL | Database and Identifier |
|---|---|---|---|---|
| Maculins T, Verschueren E, Hinkle T, Choi M, Chang P, Chalouni C, Rao S, Kwon Y, | 2020 | Multiplexed profiling of innate immune antimicrobial response - Global Proteome | https://sld-acs-sf.sf.perkinelmercloud.com/spotfire/wp/analysis?file=/Guest/OBJ0038812_Shigella_GP_msstats_ | Perkin Elmer Cloud , Shigella_GP |

| Author(s) | Year | Title | URL | Database and Identifier |
|---|---|---|---|---|
| Lim J, Katakam AK, Kunz RC, Erickson BK, Huang T, Tsai TH, Vitek O, Reichelt M, Senbabaoglu Y, Mckenzie B, Rohde JR, Dikic I, Kirkpatrick DS, Murthy A | | | tmt&waid=rQ5wpeaftEu-LyaU7_JN56-201250ef41XBf8&wavid=0 | |
| Maculins T, Verschueren E, Hinkle T, Choi M, Chang P, Chalouni C, Rao S, Kwon Y, Lim J, Katakam AK, Kunz RC, Erickson BK, Huang T, Tsai TH, Vitek O, Reichelt M, Senbabaoglu Y, Mckenzie B, Rohde JR, Dikic I, Kirkpatrick DS, Murthy A | 2020 | Multiplexed profiling of innate immune antimicrobial response - Phosphorylation | https://sld-acs-sf.sf.perki-nelmercloud.com/spot-fire/wp/analysis?file=/Guest/OBJ0038812_Shi-gella_pSTY_msstat_tmt&waid=tyfGYhcgZ-ka083K3Ud3yG-201250ef41XBf8&wavid=0 | Perkin Elmer Cloud , Shigella_pSTY |
| Maculins T, Verschueren E, Hinkle T, Choi M, Chang P, Chalouni C, Rao S, Kwon Y, Lim J, Katakam AK, Kunz RC, Erickson BK, Huang T, Tsai TH | 2020 | Multiplexed profiling of innate immune antimicrobial response - Ubiquitin | https://sld-acs-sf.sf.perki-nelmercloud.com/spot-fire/wp/analysis?file=/Guest/OBJ0038812_Shi-gella_KGG_msstat_tmt&waid=uw0mWL2K-GEOkBjieRqKRv-201250ef41XBf8&wavid=0 | Listed in data availability statement, Shigella_KGG |
| Maculins T, Verschueren E, Hinkle T, Choi M, Chang P, Chalouni C, Rao S, Kwon Y, Lim J, Katakam AK, Kunz RC, Erickson BK, Huang T, Tsai TH, Vitek O, Reichelt M, Senbabaoglu Y, Mckenzie B, Rohde JR, Dikic I, Kirkpatrick DS, Murthy A | 2021 | Multiplexed proteomics of autophagy-deficient murine macrophages reveals enhanced antimicrobial immunity via the oxidative stress response | https://massive.ucsd.edu/ProteoSAFe/data-set.jsp?accession=MSV000085565 | UCSD MassIVE, MSV000085565 |
| Maculins T, Verschueren E, Hinkle T, Choi M, Chang P, Chalouni C, Rao S, Kwon Y, Lim J, Katakam AK, Kunz RC, Erickson BK, Huang T, Tsai TH, Vitek O, Reichelt M, Senbabaoglu Y, Mckenzie B, Rohde JR, Dikic I, Kirkpatrick DS, Murthy A | 2021 | Multiplexed proteomics of autophagy-deficient murine macrophages reveals enhanced antimicrobial immunity via the oxidative stress response | https://sourceforge.net/projects/sashimi/files/ReAdW%20%28Xcalibur%20converter%29/ | Sourceforge.net, ReAdW%20%28Xcalibur%20converter%29/ |
| Maculins T, Verschueren E, Hinkle T, Choi M, Chang P, Chalouni C, Rao S, Kwon Y, Lim J, Katakam AK, Kunz RC, Erickson | 2021 | Multiplexed proteomics of autophagy-deficient murine macrophages reveals enhanced antimicrobial immunity via the oxidative stress response | https://www.bioconduc-tor.org/packages/re-lease/bioc/html/MSstatsTMT.html | Bioconductor , 10.18129/B9.bioc.MSstatsTMT |

BK, Huang T, Tsai
TH, Vitek O,
Reichelt M,
Senbabaoglu Y,
Mckenzie B, Rohde
JR, Dikic I,
Kirkpatrick DS,
Murthy A

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
