## [Decision Letter]

**Acceptance summary:**

In this manuscript, the authors examine the role of ATG16L1 in macrophages in controlling *Shigella flexneri* infection. The authors report an interesting link between regulation of oxidative stress by autophagy genes and Shigella pathogenesis. These findings reveal important information regarding this clinically relevant pathogen as well as provide important resources for the field.

**Decision letter after peer review:**

Thank you for submitting your article "Proteomics of autophagy deficient macrophages reveals enhanced antimicrobial immunity via the oxidative stress response" for consideration by *eLife*. Your article has been reviewed by 3 peer reviewers, one of whom is a member of our Board of Reviewing Editors, and the evaluation has been overseen by Wendy Garrett as the Senior Editor. The reviewers have opted to remain anonymous.

The reviewers have discussed the reviews with one another and the Reviewing Editor has drafted this decision to help you prepare a revised submission.

Summary:

In this manuscript, the authors examine the role of ATG16L1 in macrophages in controlling *Shigella flexneri* infection. The authors report that: (1) macrophages deficient in ATG16L1 demonstrate accelerated killing of *Shigella flexneri in vitro*, (2) mice harboring myeloid ATG16L1 deficiency control Shigella better *in vivo*, (3) a panel of changes in global proteome and post-translational modifications occur, pointing to an accumulation of ROS in ATG16L-deficient BMDM, and (4) depletion of or reduced but not oxidized GSH in cells with Erastin accelerates Shigella clearance by WT macrophages.

The authors have also created a website for users to investigate the proteomics data in detail, providing an important resource to the community.

The discovery of a new mechanism by which autophagy proteins affect host defenses would be very exciting. However, although the authors present a compelling model of how Atg16L1 promotes Shigella survival based on their proteomics data; the manuscript currently falls short of proving the model. Although the authors show increased ROS in Atg16L1-deficient macrophages, they do not present evidence to show this is responsible for the modest decrease in CFUs *in vitro* and whether any of the *in vitro* phenotypes directly relate to the *in vivo* decrease in liver burden in the absence of ATG16L1 in myeloid cells.

Thus, there are still many critical questions that remain unanswered. These critical questions are as follows. Does ATG16L suppress Shigella clearance in macrophages through regulating ROS *in vivo*? Does this role of ATG16L involve the process of autophagy? In addition, the authors make a point of emphasizing the strengths of their proteomic approach in identifying downstream mechanisms; however, this aspect of the manuscript requires further verification and detailed analysis.

Essential revisions:

Revisions required to connect ROS production in macrophages to control of Shigella replication *in vivo*:

1. The authors report that Atg16L1-cKO mice display reduced shigella liver burden and conclude that macrophages deficient in ATG16L1 demonstrate an accelerated killing of Shigella *in vivo*. However, LysM-Cre mediates deletion in a wide range of myeloid cells systemically in the mouse. Is the effect the authors reported specific to the liver? Is this mediated by Atg16L1-deficient Kupffer cells in the liver?

2. There are numerous ways autophagy proteins affect inflammatory responses (as also observed in the authors' proteomic analysis) and, therefore, the *in vivo* phenotype may be due to inflammatory differences and not due to a direct role for Atg16L1 in a cell-intrinsic antimicrobial pathway in macrophages. The authors present evidence that several cytokine pathways are significantly affected by loss of Atg16l1, including interferons and TNF. Since autophagy affects cytokine production and release, is it possible that increased ROS and pathogen clearance is a consequence of increased cytokine signaling in the basal state and after infection? This could also explain the observed changes in protein abundance or modifications.

3. For the Erastin data, this was performed in WT BMDMs, not ATG16L1 deficient BMDMs, and therefore, may have nothing to do with what is going on in ATG16L1 cKOs. The authors will need to show directly that increased ROS in ATG16L1 cKOs is leading to decreased bacterial replication. In addition, the authors do not demonstrate that the Erastin treatment resulted in elevated ROS. Notably, the difference in CFUs following Erastin treatment was much larger in magnitude than that reported for loss of ATG16L1. Another helpful approach here may be the addition of a ROS scavenger (e.g. NAC) in experimental conditions to functionally test the role of ROS in the observed phenotypes.

4. There appears to be significant variability in the GSSG/GSH ratio in sFigure 7a. Is there a more reproducible or comprehensive way to measure the redox state? Perhaps by additionally examining protein carbonyl adduct formation or other forms of damage (lipids, DNA)?

5. As a result of the absence of linkage between the authors' observations, there are many times that data are over-interpreted. For example, in the last line of abstract: "These findings demonstrate that control of oxidative stress by ATG16L1 regulates antimicrobial immunity against intracellular pathogens". The authors never show that control of oxidative stress by ATG16L1 regulates antimicrobial immunity. Another example is in Lines 132-135- "Our findings demonstrate that ATG16L1 tunes antimicrobial immunity against cytosolic pathogens via the oxidative stress response…". This is not supported by the data presented. The authors also start a number of sentences with the "Interestingly" or "Unexpectedly" even when the data could be expected. For example, Line 310, loss of autophagy has already been linked to increased ROS in other systems. Please address these issues with text revision.

Revisions required to evaluate and verify the proteomics data:

1. To examine potential mechanisms, the authors performed global proteomic analysis in BMDMs treated ex vivo with Shigella. This led to the identification of numerous modifications in phosphorylation, ubiquitylation and protein abundance. The authors focused on autophagy cargo adaptors, and described numerous modifications to TAX1BP1, OPTN, p62, etc. The proteomic analysis was performed by tandem mass tagging, which allowed comparisons across 2 11-plex experiments. The authors provide a detailed overview of modifications of proteins in the autophagy pathway, including substrate-specific receptors. However, there are a few limitations of the study: (1) There isn't much in the way of comparisons with phosphorylation and ubiquitylation patterns seen previously when these proteins – e.g. TAX1BP1 or OPTN – are phosphorylated during selective forms of autophagy. This has been examined in multiple papers previously (e.g. PMID: 26365381; PMID: 27035970; PMID: 25972374 ).

2. Although all the modification data is nice and provides an overview, the authors don't really use it for analysis of mechanisms. Are any of the phosphorylation or ubiquitylation events central to the response to Shigella or loss of ATG16L1? As presented, the modification side of the story doesn't go anywhere mechanistically.

3. The stoichiometry of the modifications are not addressed. While some modifications appear to have pretty large fold changes, many are only ~2-fold, so it may be that this is a tiny fraction of the protein that is modified. Therefore, it is hard to know what is significant biologically without further follow-up. For example, is the modification of the cargo adaptors linked with removal of Shigella directly or is it some other role (for example turnover of aggregated proteins as a result of increased ROS). For example, TAX1BP1 T494 phosphorylation is already maximal in the ATG16L1 KO without Shigella (Figure 3). Similarly, the relationship between phosphorylation and ubiquitylation on the same polypeptide (i.e. combinatorial modification) isn't addressed.

4. The e3 ubiquitin ligase and xenobiotic metabolism pathways are increased in ATG16L1 KO macrophages. This is suggestive of increased aryl hydrocarbon receptor activity. Since AhR is a driver of ROS in the cell, as part of the xenobiotic detoxifying response, it would be interesting to see if AhR was involved in the observed phenotype.

---

## [Author Response]

Essential revisions:Revisions required to connect ROS production in macrophages to control of Shigella replication *in vivo*:1. The authors report that Atg16L1-cKO mice display reduced shigella liver burden and conclude that macrophages deficient in ATG16L1 demonstrate an accelerated killing of Shigella *in vivo*. However, LysM-Cre mediates deletion in a wide range of myeloid cells systemically in the mouse. Is the effect the authors reported specific to the liver? Is this mediated by Atg16L1-deficient Kupffer cells in the liver?Is the effect the authors reported specific to the liver?

*S.flexneri* is a human enteric pathogen with poor tropism for murine intestinal tissue. As such, it is unable to colonize the intestinal mucosa in mice unless genetically modified strains are utilized that compromise inflammasome function in the intestinal epithelium (recently described by Mitchell et al., 2020). Thus, infection via the tail-vein is commonly used as an acute model to approximate bacterial colonization *in vivo*. We have added this clarification in the text to better outline the limitations of current *in vivo* models (Page 7; Lines 176-179 and Page 17; Lines 495-500).

In response to this query, we provide new data showing comparison of additional tissues beyond the liver. Specifically, we observe that infection of the spleen is comparably low, without a significant difference between *ATG16L1-WT* (control) and *ATG16L1-cKO* animals. This data is presented in new panel Figure S1B. Additionally, we evaluated lung tissue and found that this tissue does not show detectable infection (data not shown) and added a comment about this in the text (Page 7, 8; Lines 181-185).

Is this mediated by Atg16L1-deficient Kupffer cells in the liver?

We agree that the *in vivo* model (*LysM*-Cre) utilized in the manuscript does not formally demonstrate Kupffer cells as the sole driver of the phenotype, however, the model permits studies with targeted gene deletion in the monocyte and Kupffer cell populations. We have added clarifications in the manuscript text to acknowledge this limitation (Page 8; Lines 185-193. Pages 17, 18; Lines 503-508).

We agree that the *LysM-*Cre model also edits granulocyte populations which may impact the phenotype at later time points following liver injury. We have adjusted our interpretation of the data to reflect this limitation. Additionally, to better focus our findings on macrophage populations, we repeated the infection studies and quantified bacterial colonization at an earlier infection time point (6 hours post infection). This enabled us to observe that mice lacking Atg16l1 in the myeloid compartment (*LysM*-Cre) continue to exhibit significantly decreased *S.flexneri* CFU in the liver. This data is presented in new panel Figure 1D.

We improved the manuscript text accordingly and hope this addresses the Reviewer concern.

2. There are numerous ways autophagy proteins affect inflammatory responses (as also observed in the authors' proteomic analysis) and, therefore, the *in vivo* phenotype may be due to inflammatory differences and not due to a direct role for Atg16L1 in a cell-intrinsic antimicrobial pathway in macrophages. The authors present evidence that several cytokine pathways are significantly affected by loss of Atg16l1, including interferons and TNF. Since autophagy affects cytokine production and release, is it possible that increased ROS and pathogen clearance is a consequence of increased cytokine signaling in the basal state and after infection? This could also explain the observed changes in protein abundance or modifications...is it possible that increased ROS and pathogen clearance is a consequence of increased cytokine signaling in the basal state and after infection?

Thank you for this important question. Indeed, our quantitative mass spectrometry data (new Figure 5—figure supplement 1) and RNA-Seq data (new panels Figure 5—figure supplement 2A and B) demonstrate a significant enrichment of pro-inflammatory signaling pathways either at baseline or following infection of cKO BMDMs. We addressed the role of TNFa and Type I Interferon signaling in pathogen clearance using BMDM infection model. Indeed, blocking TNFa or IFNAR1 resulted in a modest but significant rescue of bacterial proliferation in cKO, but not WT, BMDMs (new data in Figure 5—figure supplement 2C-H). This is in contrast to modulation of ROS levels that affect bacterial proliferation in both WT and cKO BMDMs (new Figure 6). Therefore, it is likely that contributions from pro-inflammatory signaling and ROS together drive bacterial clearance in the BMDM infection model, with a more pronounced phenotype in cells lacking ATG16L1.

We then decided to follow this observation *in vivo* and performed a study aiming to assess the role of overactive interferon signaling inherent to ATG16L1 KO animals in clearing *S.flexneri*. We focused on blocking type I IFN signaling given the more significant impact of a-IFNAR1 treatment on cKO BMDMs (new panel Figure 5—figure supplement 2G). Briefly, mice were pre-treated via intraperitoneal injections (IP) with isotype control or anti-IFNAR1 antibodies for 24h. We used anti-IFNAR1 antibody clone MAR1-5A3 2.5 mg per mouse (Leinco Technologies). Mice were injected with antibodies again prior infection (0.5 mg per mouse, IP) and infected with *S.flexneri* as before (IV). Subsequently, 24h after infection the livers were collected for CFU analysis.

The result shown in Author response image 1 demonstrates a significant CFU reduction in livers from ATG16L1-cKO animals in the isotype control group, which is consistent with our observations presented in panels Figure 1D and 1E. We also observed a trend towards higher CFU counts in ATG16L1-cKO animals treated with the anti-IFNAR1 antibody. However, this experiment failed to demonstrate a statistically significant increase in CFUs within each genotype. *S.flexneri* CFUs did not increase significantly between isotype-treated and anti-IFNAR1-treated WT or cKO animals. In addition to the inherent variability of this challenging model, other factors could contribute to the lack of a clear rescue of bacterial proliferation. These include an incomplete pharmacological blockade of IFNAR1 signaling and redundant pathways which control bacterial proliferation. Additional genetic models (e.g. combined deletion of *Atg16l1* and *Ifnar1*) or combination therapies are currently out-of-scope for the study but supported by our findings.

Taken together, experiments to address this query led us to acknowledge a contribution of pro-inflammatory signaling in clearing *S.flexneri* in our *in vitro* BMDM infection model and introduce a new Figure S8 to reflect this aspect. We thank the Reviewer for this important question.

**Author response image 1. respfig1:** *in vivo* blockade of Type I Interferon signaling during *S.flexneri* infection.

3. For the Erastin data, this was performed in WT BMDMs, not ATG16L1 deficient BMDMs, and therefore, may have nothing to do with what is going on in ATG16L1 cKOs.The authors will need to show directly that increased ROS in ATG16L1 cKOs is leading to decreased bacterial replication.

We addressed this experimentally and by determining whether Erastin treatment of ATG16L1-deficient BMDMs further accelerated *S.flexneri* clearance. Indeed, this was the case, which is aligned with the main message of our manuscript. We thank Reviewer for this suggestion. This data is presented in new panels Figure 6B and 6C.

In addition, the authors do not demonstrate that the Erastin treatment resulted in elevated ROS.

In the initial submission, we provided data showing that Erastin treatment for 24 hours resulted in elevated oxidative stress, prior to evaluating its effects on bacterial clearance (former panel Supplemental Figure 7G). While WT BMDMs exhibited a clear increase in oxidative stress upon Erastin treatment, ATG16L1-deficient BMDMs *with basally elevated oxidative stress* did not show a further increase at this time point. We also observed that prolonged Erastin treatment resulted in BMDM cytotoxicity, with ATG16L1-deficient BMDMs exhibiting enhanced sensitivity to the inhibitor at a later time point (new Figure 6—figure supplement 1A). This is consistent with our findings that ATG16L1-deficient BMDMs are dependent on XCT/GSH mediated antioxidant pathways. It is technically challenging to measure oxidative stress at a later time-point since a large population of ATG16L1-deficient BMDMs are undergoing cell death. However, following Reviewer recommendations, we measured oxidative stress upon prolonged Erastin treatment over 48h. We used high content microscopy enabling imaging of larger cell populations. Consistent with our previous observations, we detected a highly significant increase in CellRox probe mean intensity between untreated WT and cKO BMDMs (this data is shown here as Author response image 2 and presented in a new panel Figure 5—figure supplement 3A). Consistent with the established role of Erastin in increasing oxidative stress, we also observed a significant increase in CellRox probe mean intensity following 48h treatment with Erastin. We hope that the data presented here addresses the Reviewer question. Taken together, we decided to remove the initial data presented in the first submission in relation to this question (former panel Supplemental Figure 7G).

**Author response image 2. respfig2:** Quantification of CellRox green mean intensity in WT and cKO BMDMs. Graph shows single cell data from high content imaging (n = 1). Unpaired t test **** P < 0.0001.

Notably, the difference in CFUs following Erastin treatment was much larger in magnitude than that reported for loss of ATG16L1

Thank you for this question. During optimization experiments we observed that the larger difference in CFUs is a reflection of drug exposure. A dose-titration of Erastin revealed a dose-dependent decrease in bacterial CFUs. In Author response image 3 we provide a dose-titration of Erastin and its impact on *S.flexneri* CFUs in response to Reviewer question. We hope that this answers Reviewer question.

**Author response image 3. respfig3:** Dose-titration of Erastin and its impact on *S.flexneri* killing by BMDMs.

Another helpful approach here may be the addition of a ROS scavenger (e.g. NAC) in experimental conditions to functionally test the role of ROS in the observed phenotypes.

We thank the Reviewer for this important suggestion. We assessed if a ROS scavenger (butylated hydroxyanisole or BHA) is capable to rescue *S.flexneri* killing in our experimental *in vitro* infection system using both WT and ATG16L1-deficient macrophages. We found that BHA was able to rescue bacterial killing by both WT and cKO BMDMs. This data is now a part of new Figure 6.

4. There appears to be significant variability in the GSSG/GSH ratio in sFigure 7a. Is there a more reproducible or comprehensive way to measure the redox state? Perhaps by additionally examining protein carbonyl adduct formation or other forms of damage (lipids, DNA)?

We believe there are several reasons for variable results presented in the Supplemental Figure 9B-D of the resubmitted revised manuscript. We noted that only a subpopulation of ATG16L1-deficient BMDMs demonstrates increased cellular ROS levels in our single-cell imaging data presented in Figure 5E. We have followed up on this observation using high content single-cell imaging for cellular ROS and obtained very similar data for a large population of cells (>1000 cells per genotype). We believe this stochasticity is an important point to convey and provide the high content imaging data to the supplement of the revised manuscript (new panel Figure 5—figure supplement 3A).

The GSSG/GSH ELISA-type experiments presented in Figure 5—figure supplement 3B-D, to which the Reviewer refers, are performed using cell lysate from a population of cells, which likely accounts for some of this variability since it fails to achieve single-cell resolution as in the above assays. An additional factor contributing to this variability is the exclusive usage of primary macrophages in this study, which lends itself to variability between independent experiments. Nonetheless, we believe the data presented in Figure 5—figure supplement 3B-D show a consistent trend across independent biological repeats of the assay.

Additionally, we now provide further evidence to support our observations. In our initial submission, we noted that ATG16L1 deficient BMDMs exhibit elevated levels of the NAD(P)H:quinone acceptor oxidoreductase NQO1 (Figure 5B). Elevated NQO1 levels and activity are indicative of a heightened redox state (Reviewed by Ross et al., 2017; Siegel et al., 2018; Raina et al., 1999). To address the Reviewer’s query, we have synthesized a recently described NQO1-activated fluorescent probe (Punganuru et al., 2019), which enabled us to assess the redox state of WT and ATG16L1-deficient BMDMs using flow cytometry. These experiments represent a highly sensitive assay to measure redox state at the single-cell level in primary cells and demonstrate and increased MFI of the probe in cKO BMDMs. This data constitute new panels Figure 5—figure supplement 3E and F.

We have also attempted quantification of s-GSH adducts and 4-HNE damaged lipids by competitive ELISA kits, however were not able to achieve reliable assay performance using assay standards, which is an essential step prior evaluating assay results (data not shown).

We hope the Reviewer will find our answer and the provided data satisfactory.

5. As a result of the absence of linkage between the authors' observations, there are many times that data are over-interpreted. For example, in the last line of abstract: "These findings demonstrate that control of oxidative stress by ATG16L1 regulates antimicrobial immunity against intracellular pathogens". The authors never show that control of oxidative stress by ATG16L1 regulates antimicrobial immunity. Another example is in Lines 132-135- "Our findings demonstrate that ATG16L1 tunes antimicrobial immunity against cytosolic pathogens via the oxidative stress response…". This is not supported by the data presented. The authors also start a number of sentences with the "Interestingly" or "Unexpectedly" even when the data could be expected. For example, Line 310, loss of autophagy has already been linked to increased ROS in other systems. Please address these issues with text revision.

We hope that new data generated with the aim of addressing previously discussed queries (Questions 1-4) also address the concerns above. Specifically:

1. We show elevated oxidative stress and downstream responses in ATG16L1 cKO BMDMs via new assays additional high content imaging of cellular ROS – new Figure 5—figure supplement 3A; elevated NQO1 activity via a new fluorescent probe – new Figure 5—figure supplement 3E, F; additional RNA-Seq analysis – new Figure 5—figure supplement 2A for GSEA ; (2) We now show that anti-oxidant treatment (BHA) reverses the phenotype in ATG16L1 cKO BMDMs more dramatically than TNF or IFN inhibition (BHA inhibition – new Figure 6D-F, Figure 6—figure supplement 1E, F; TNF inhibition – new Figure 5—figure supplement 2C-E; IFNAR1 inhibition – new Figure 5—figure supplement 2F-H); (3) we show that XCT inhibition phenocopies enhanced anti-microbial immunity via the GSH pathway, and that ATG16L1 deficient macrophages exhibit increased dependency on XCT for viability (new Figure 6A-C, Figure 6—figure supplement 1A, C, D). We thank the Reviewers for guiding us towards these critical functional studies and believe that assertions made in the text are sufficiently supported by these new data.

We have also edited the text of the revised manuscript to avoid usage of qualifying adjectives and instead directly state the observations. We thank the Reviewer for highlighting this issue.

Revisions required to evaluate and verify the proteomics data:1. To examine potential mechanisms, the authors performed global proteomic analysis in BMDMs treated ex vivo with Shigella. This led to the identification of numerous modifications in phosphorylation, ubiquitylation and protein abundance. The authors focused on autophagy cargo adaptors, and described numerous modifications to TAX1BP1, OPTN, p62, etc. The proteomic analysis was performed by tandem mass tagging, which allowed comparisons across 2 11-plex experiments. The authors provide a detailed overview of modifications of proteins in the autophagy pathway, including substrate-specific receptors. However, there are a few limitations of the study: (1) There isn't much in the way of comparisons with phosphorylation and ubiquitylation patterns seen previously when these proteins – e.g. TAX1BP1 or OPTN – are phosphorylated during selective forms of autophagy. This has been examined in multiple papers previously (e.g. PMID: 26365381; PMID: 27035970; PMID: 25972374 ).

We are unsure of the specific question being posed here, but have revised the text to include additional commentary about signaling via autophagy pathway components, including the cargo receptors based on in the papers referenced above. The phosphorylation sites reported in these three papers (OPTN_HUMAN_S473, OPTN_HUMAN_S513, and SQSTM_HUMAN_S403) have conserved sites in MOUSE (OPTN_MOUSE_S476, OPTN_MOUSE_S517, SQSTM_MOUSE_S405), but were not identified or quantified in this work.

The heatmaps in Figure 3 show PTMs from multiple autophagy-related proteins that were identified and quantified and have been described in the text. Additionally, the interactive dashboards are provided to facilitate comparisons like the ones suggested in the question. However, the totality of the insights generated by multiplexed TMT-MS revealed altered oxidative stress response as a fundamentally important aspect of anti-microbial immunity.

2. Although all the modification data is nice and provides an overview, the authors don't really use it for analysis of mechanisms. Are any of the phosphorylation or ubiquitylation events central to the response to Shigella or loss of ATG16L1? As presented, the modification side of the story doesn't go anywhere mechanistically.

We thank the Reviewer for this question. We agree that our initial manuscript did not sufficiently highlight how the PTM data contributed to our conclusions and have tried to remedy that in this revision. Our initial observation that basal mitochondrial function was largely intact in ATG16L1-deficient cells (new Figure 5—figure supplement 3G-H) was surprising and prompted us to query additional organelle proteomes in the TMT-MS modification dataset. We focused on mitochondria and peroxisomes, given the established role of these organelles with modulation of cellular ROS. While we did not detect significant or broad ranging accumulation of mitochondrial or peroxisomal proteins, a specific enrichment of peroxisomal matrix proteins in the ubiquitinated fraction was observed in ATG16L1-deficient BMDMs, as shown in the GSEA heatmap presented in the new Figure 5—figure supplement 1C. An additional hit in the PTM dataset that points us to the intersection of oxidative stress and the peroxisome is elevated ubiquitination of Fis1 at lysine-20 (FIS1_K20). Fis1 is known to play a key role in peroxisomal homeostasis. From the phosphoproteomics data, ubiquitin phosphorylation at serine-57 also stands out given its recent connection to oxidative stress signaling in yeast (Hepowit et al., 2020).

These findings leverage the modification (PTM) data generated by TMT-MS and reveal that elevated basal oxidative stress upon Atg16l1 loss appears more closely connected to peroxisomal rather than mitochondrial dysfunction. This commentary has been included in the revised manuscript (Page 15; Lines 428-431). Additional follow up work will be required to elucidate the full mechanistic picture connecting deficient autophagy to the peroxisomal and oxidative stress pathways, but remain outside the scope of our current submission

3. The stoichiometry of the modifications are not addressed. While some modifications appear to have pretty large fold changes, many are only ~2-fold, so it may be that this is a tiny fraction of the protein that is modified. Therefore, it is hard to know what is significant biologically without further follow-up. For example, is the modification of the cargo adaptors linked with removal of Shigella directly or is it some other role (for example turnover of aggregated proteins as a result of increased ROS). For example, TAX1BP1 T494 phosphorylation is already maximal in the ATG16L1 KO without Shigella (Figure 3).

In the context of MS proteomics technologies, TMT and the majority of other approaches report relative rather than absolute quantitative values. Moreover, the majority of physiologically meaningful PTM level changes occur at sub-stoichiometric levels, owing to subcellular compartmentalization and cellular heterogeneity in the responses. Single cell profiling technologies consistently show that a physiologically relevant experimental perturbation like infection causes a diverse range of responses across a population of cells, especially when considering that individual cells are infected at different rates both *in vivo* and ex vivo. In contrast, non-physiological biochemical treatments such as high level LPS or CpG stimulation show more consistent, population-wide changes but do not faithfully recapitulate the infection cycle of a pathogen.

Similarly, the relationship between phosphorylation and ubiquitylation on the same polypeptide (i.e. combinatorial modification) isn't addressed.

Given the transient nature of most ubiquitination events, the low stoichiometries of both ubiquitination and phosphorylation sites within the population, and the distribution of these post-translational modifications across the proteome, the frequency doubly modified peptides bearing these two heterogenous marks is exceedingly rare in this and other proteomics data. This can be seen within the existing data when looking at the relative (in)frequency of multiply phosphorylated or multiply ubiquitinated peptides, compared to singly modified peptides.

It is important to call out an analytical caveat that additionally hampers this investigation. Permuting an all by all matrix of two post-translational modifications at proteome scale results in a combinatorial explosion that poses challenges for the software tools tasked with identifying and quality filtering these rare peptide spectral matches.

4. The e3 ubiquitin ligase and xenobiotic metabolism pathways are increased in ATG16L1 KO macrophages. This is suggestive of increased aryl hydrocarbon receptor activity. Since AhR is a driver of ROS in the cell, as part of the xenobiotic detoxifying response, it would be interesting to see if AhR was involved in the observed phenotype.

During the revision we took a deeper look by performing GSEA with larger curated MSigDb gene set collections to see if there was an indication of AhR pathway that might further explain the quantitative changes in *S.flexneri* infected and ATG16L1-deficient macrophages. Unfortunately, no evidence was found that points to a connection of this *S. flexneri* infection of WT or cKO BMDMs directly to the AhR signaling pathway.